# Unsupervised Foreground Extraction via Deep Region Competition

**Peiyu Yu**[1]
yupeiyu98@cs.ucla.edu

**Sirui Xie**[1]
srxie@ucla.edu

**Xiaojian Ma**[1]
xiaojian.ma@ucla.edu

**Yixin Zhu**[3]
y@bigai.ai

**Ying Nian Wu**[2]
ywu@stat.ucla.edu

**Song-Chun Zhu**[1,2,3]
sczhu@stat.ucla.edu

[1]UCLA Department of Computer Science    [2]UCLA Department of Statistics
[3]Beijing Institute for General Artificial Intelligence (BIGAI)

## Abstract

We present Deep Region Competition (DRC), an algorithm designed to extract foreground objects from images in a fully unsupervised manner. Foreground extraction can be viewed as a special case of generic image segmentation that focuses on identifying and disentangling objects from the background. In this work, we rethink the foreground extraction by reconciling energy-based prior with generative image modeling in the form of Mixture of Experts (MoE), where we further introduce the learned pixel re-assignment as the essential inductive bias to capture the regularities of background regions. With this modeling, the foreground-background partition can be naturally found through Expectation-Maximization (EM). We show that the proposed method effectively exploits the interaction between the mixture components during the partitioning process, which closely connects to region competition [1], a seminal approach for generic image segmentation. Experiments demonstrate that DRC exhibits more competitive performances on complex real-world data and challenging multi-object scenes compared with prior methods. Moreover, we show empirically that DRC can potentially generalize to novel foreground objects even from categories unseen during training.[1]

## 1   Introduction

Foreground extraction, being a special case of generic image segmentation, aims for a binary partition of the given image with specific semantic meaning, *i.e.*, a foreground that typically contains identifiable objects and the possibly less structural remaining regions as the background. There is a rich literature on explicitly modeling and representing a given image as foreground and background (or more general visual regions), such that a generic inference algorithm can produce plausible segmentations ideally for any images without or with little supervision [1–8]. However, such methods essentially rely on low-level visual features (*e.g.*, edges, color, and texture), and some further require human intervention at initialization [4, 5], which largely limits their practical performance on modern datasets of complex natural images with rich semantic meanings [9, 10]. These datasets typically come with fine-grained semantic annotations, exploited by supervised methods that learn representation and inference algorithm as one monolithic network [11–16]. Despite the success of densely supervised learning, the unsupervised counterpart is still favored due to its resemblance to how humans perceive the world [17, 18].

---

[1]Code and data available at https://github.com/yuPeiyu98/DRC

35th Conference on Neural Information Processing Systems (NeurIPS 2021).

Attempting to combine unsupervised or weakly supervised learning with modern neural networks, three lines of work surge recently for foreground extraction: (1) deep networks as feature extractors for canonical segmentation algorithms, (2) GAN-based foreground-background disentanglement, and (3) compositional latent variable models with slot-based object modeling. Despite great successes of these methods, the challenge of unsupervised foreground extraction remains largely open.

Specifically, the first line of work trains designated deep feature extractors for canonical segmentation algorithms or metric networks as learned partitioning criteria [19–21]. These methods (*e.g.*, W-Net [19]) define foreground objects' properties using learned features or criteria and are thus generally bottle-necked by the selected post-processing segmentation algorithm [22, 23]. As a branch of pioneering work that moves beyond these limitations, Yang et al. [24, 25] have recently proposed a general contextual information separation principle and an efficient adversarial learning method that is generally applicable to unsupervised segmentation, separation and detection. GAN-based models [26–31] capture the foreground objectness with oversimplified assumptions or require additional supervision to achieve foreground-background disentanglement. For example, the segmentation model in ReDO [28] is trained by redrawing detected objects, which potentially limits its application to datasets with diverse object shapes. OneGAN [31] and its predecessors [29, 30], though producing impressive results on foreground extraction, require a set of background images without foreground objects as additional inputs. Lastly, compositional latent variable models [32–40] include the background as a "virtual object" and induce the independence of object representations using an identical generator for all object slots. Although these methods exhibit strong performance on synthetic multi-object datasets with simple backgrounds and foreground shapes, they may fail on complex real-world data or even synthetic datasets with more challenging backgrounds [37, 38]. In addition, few unsupervised learning methods have provided explicit identification of foreground objects and background regions. While they can generate valid segmentation masks, most of these methods do not specify which output corresponds to the foreground objects. These deficiencies necessitate rethinking the problem of unsupervised foreground extraction. We propose to confront the challenges in formulating (1) a generic inductive bias for modeling foreground and background regions that can be baked into neural generators, and (2) an effective inference algorithm based on a principled criterion for foreground-background partition.

Inspired by Region Competition [1], a seminal approach that combines optimization-based inference [41–43] and probabilistic visual modeling [44, 45] by minimizing a generalized Bayes criterion [46], we propose to solve the foreground extraction problem by reconciling energy-based prior [47] with generative image modeling in the form of Mixture of Experts (MoE) [48, 49]. To generically describe background regions, we further introduce the learned pixel re-assignment as the essential inductive bias to capture their regularities. Fueled by our modeling, we propose to find the foreground-background partition through Expectation-Maximization (EM). Our algorithm effectively exploits the interaction between the mixture components during the partitioning process, echoing the intuition described in Region Competition [1]. We therefore coin our method Deep Region Competition (DRC). We summarize our **contributions** as follows:

1. We provide probabilistic foreground-background modeling by reconciling energy-based prior with generative image modeling in the form of MoE. With this modeling, the foreground-background partition can be naturally produced through EM. We further introduce an inductive bias, *pixel re-assignment*, to facilitate foreground-background disentanglement.

2. In experiments, we demonstrate that DRC exhibits more competitive performances on complex real-world data and challenging multi-object scenes compared with prior methods. Furthermore, we empirically show that using learned pixel re-assignment as the inductive bias helps to provide explicit identification for foreground and background regions.

3. We find that DRC can potentially generalize to novel foreground objects even from categories unseen during training, which may provide some inspiration for the study of out-of-distribution (OOD) generalization in more general unsupervised disentanglement.

## 2 Related Work

A typical line of methods frames unsupervised or weakly supervised foreground segmentation within a generative modeling context. Several methods build upon generative adversarial networks (GAN) [26] to perform foreground segmentation. LR-GAN [27] learns to generate background re-

gions and foreground objects separately and recursively, which simultaneously produces the foreground objects mask. ReDO (ReDrawing of Objects) [28] proposes a GAN-based object segmentation model, based on the assumption that replacing the foreground object in the image with a generated one does not alter the distribution of the training data, given that the foreground object is correctly discovered. Similarly, SEIGAN [29] learns to extract foreground objects by recombining the foreground objects with the generated background regions. FineGAN [30] hierarchically generates images (*i.e.*, first specifying the object shape and then the object texture) to disentangle the background and foreground object. Benny and Wolf [31] further hypothesize that a method solving an ensemble of unsupervised tasks altogether improves the model performance compared with the one that solves each individually. Therefore, they train a complex GAN-based model (OneGAN) to solve several tasks simultaneously, including foreground segmentation. Although LR-GAN and FineGAN do produce masks as part of their generative process, they cannot segment a given image. Despite SEIGAN and OneGAN achieving decent performance on foreground-background segmentation, these methods require a set of clean background images as additional inputs for weak supervision. ReDO captures the foreground objectness with possibly oversimplified assumptions, limiting its application to datasets with diverse object shapes.

On another front, compositional generative scene models [32–40], sharing the idea of scene decomposition stemming from DRAW [50], learn to represent foreground objects and background regions in terms of a collection of latent variables with the same representational format. These methods typically exploit the spatial mixture model for generative modeling. Specifically, IODINE [37] proposes a slot-based object representation method and models the latent space using iterative amortized inference [51]. Slot-Attention [38], as a step forward, effectively incorporates the attention mechanism into the slot-based object representation for flexible foreground object binding. Both methods use fully shared parameters among individual mixture components to entail permutation invariance of the learned multi-object representation. Alternative models such as MONet [36] and GENESIS [39] use multiple encode-decode steps for scene decomposition and foreground object extraction. Although these methods exhibit strong performance on synthetic multi-object datasets with simple background and foreground shapes, they may fail when dealing with complex real-world data or even synthetic datasets with more challenging background [37, 38].

More closely related to the classical methods, another line of work focuses on utilizing image features extracted by deep neural networks or designing energy functions based on data-driven methods to define the desired property of foreground objects. Pham et al. [52] and Silberman et al. [53] obtain impressive results when depth images are accessible in addition to conventional RGB images, while such methods are not directly applicable for data with RGB images alone. W-Net [19] extracts image features via a deep auto-encoder jointly trained by minimizing reconstruction error and normalized cut. The learned features are further processed by CRF smoothing to perform hierarchical segmentation. Kanezaki [20] proposes to employ a neural network as part of the partitioning criterion (inspired by Ulyanov et al. [54]) to minimize the chosen intra-region pixel distance for segmentation directly. Ji et al. [21] propose to use Invariant Information Clustering as the objective for segmentation, where the network is trained to be part of the learned distance. As an interesting extension, one may also consider adapting methods that automatically discover object structures [55] to foreground extraction. Though being pioneering work in image segmentation, the aforementioned methods are generally bottle-necked by the selected post-processing segmentation algorithm or require extra transformations to produce meaningful foreground segmentation masks. Yang et al. [24, 25] in their seminal work propose an information-theoretical principle and adversarial contextual model for unsupervised segmentation and detection by partitioning images into maximally independent sets, with the objective of minimizing the predictability of one set by the other sets. Additional efforts have also been devoted to weakly supervised foreground segmentation using image classification labels [56–58], bounding boxes [59, 60], or saliency maps [61–63].

## 3   Methodology

Foreground extraction performs a binary partition for the image $\mathbf{I}$ to extract the foreground region. Without explicit supervision, we propose to use learned pixel re-assignment as a generic inductive bias for background modeling, upon which we derive an EM-like partitioning algorithm. Compared with prior methods, our algorithm can handle images with more complex foreground shapes and background patterns, while providing explicit identification of foreground and background regions.

## 3.1 Preliminaries

Adopting the language of EM algorithm, we assume that for the observed sample $\mathbf{x} \in \mathbb{R}^D$, there exists $\mathbf{z} \in \mathbb{R}^d$ as its latent variables. The complete-data distribution is

$$p_\theta(\mathbf{z}, \mathbf{x}) = p_\alpha(\mathbf{z})p_\beta(\mathbf{x}|\mathbf{z}), \tag{1}$$

where $p_\alpha(\mathbf{z})$ is the prior model with parameters $\alpha$, $p_\beta(\mathbf{x}|\mathbf{z})$ is the top-down generative model with parameters $\beta$, and $\theta = (\alpha, \beta)$.

The prior model $p_\alpha(\mathbf{z})$ can be formulated as an energy-based model, which we refer to as the Latent-space Energy-Based Model (LEBM) [47] throughout the paper:

$$p_\alpha(\mathbf{z}) = \frac{1}{Z_\alpha} \exp\left(f_\alpha(\mathbf{z})\right) p_0(\mathbf{z}), \tag{2}$$

where $f_\alpha(\mathbf{z})$ can be parameterized by a neural network, $Z_\alpha$ is the partition function, and $p_0(\mathbf{z})$ is a reference distribution, assumed to be isotropic Gaussian prior commonly used for the generative model. The prior model in Eq. (2) can be interpreted as an energy-based correction or exponential tilting of the original prior distribution $p_0$.

The LEBM can be learned by Maximum Likelihood Estimation (MLE). Given a training sample $\mathbf{x}$, the learning gradient for $\alpha$ is derived as shown by Pang et al. [47],

$$\delta_\alpha(\mathbf{x}) = \mathbf{E}_{p_\theta(\mathbf{z}|\mathbf{x})}\left[\nabla_\alpha f_\alpha(\mathbf{z})\right] - \mathbf{E}_{p_\alpha(\mathbf{z})}\left[\nabla_\alpha f_\alpha(\mathbf{z})\right]. \tag{3}$$

In practice, the above expectations can be approximated by Monte-Carlo average, which requires sampling from $p_\theta(\mathbf{z}|\mathbf{x})$ and $p_\alpha(\mathbf{z})$. This step can be done with stochastic gradient-based methods, such as Langevin dynamics [64] or Hamiltonian Monte Carlo [65].

An extension to LEBM is to further couple the vector representation $\mathbf{z}$ with a symbolic representation $\mathbf{y}$ [66]. Formally, $\mathbf{y}$ is a K-dimensional one-hot vector, where $K$ is the number of possible $\mathbf{z}$ categories. Such symbol-vector duality can provide extra entries for auxiliary supervision; we will detail it in Section 3.4.

## 3.2 Generative Image Modeling

**Mixture of Experts (MoE) for Image Generation**   Inspired by the regional homogenity assumption proposed by Zhu and Yuille [1], we use separate priors and generative models for foreground and background regions, indexed as $\alpha_k$ and $\beta_k$, $k = 1, 2$, respectively; see Fig. 1. This design leads to the form of MoE [48, 49] for image modeling, as shown below.

Let us start by considering only the i-th pixel of the observed image $\mathbf{x}$, denoted as $\mathbf{x}_i$. We use a binary one-hot random variable $\mathbf{w}_i$ to indicate whether the i-th pixel belongs to the foreground region. Formally, we have $\mathbf{w}_i = [w_{i1}, w_{i2}]$, $w_{ik} \in \{0, 1\}$ and $\sum_{k=1}^2 w_{ik} = 1$. Let $w_{i1} = 1$ indicate that the i-th pixel $\mathbf{x}_i$ belongs to the foreground, and $w_{i2} = 1$ indicate the opposite.

We assume that the distribution of $\mathbf{w}_i$ is prior-dependent. Specifically, the mixture parameter $\pi_{ik}$, $k = 1, 2$, is defined as the output of a gating function $\pi_{ik} = p_\beta(w_{ik} = 1|\mathbf{z}) = \text{Softmax}(l_{ik})$; $l_{ik} = h_{\beta_k}(\mathbf{z}_k)$, $k = 1, 2$ are the logit scores given by the foreground and background generative models respectively; $\beta = \{\beta_1, \beta_2\}$, $\mathbf{z} = \{\mathbf{z}_1, \mathbf{z}_2\}$. Taken together, the joint distribution of $\mathbf{w}_i$ is

$$p_\beta(\mathbf{w}_i|\mathbf{z}) = \prod_{k=1}^2 \pi_{ik}^{w_{ik}}. \tag{4}$$

The learned distribution of foreground and background contents are

$$p_\beta(\mathbf{x}_i|w_{ik} = 1, \mathbf{z}_k) = p_{\beta_k}(\mathbf{x}_i|\mathbf{z}_k) \sim \mathbf{N}(g_{\beta_k}(\mathbf{z}_k), \sigma^2\mathbf{I}), \ k = 1, 2 \tag{5}$$

where we assume that the generative model for region content, $p_{\beta_k}(\mathbf{x}_i|\mathbf{z}_k)$, $k = 1, 2$, follows a Gaussian distribution parameterized by the generator network $g_{\beta_k}$. As in VAE, $\sigma$ takes an assumed value. We follow the common practice and use a shared generator for parameterizing $\pi_{ik}$ and $p_{\beta_k}(\mathbf{x}_i|\mathbf{z}_k)$. We use separate branches only at the output layer to generate logits and contents.

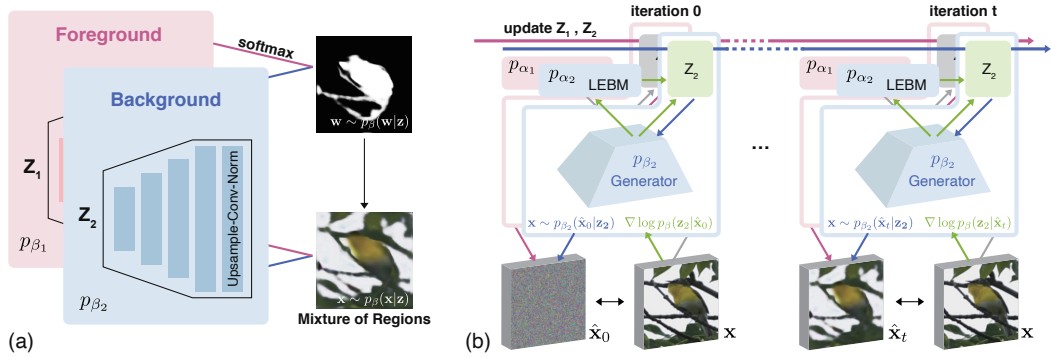

(a)

(b)

Figure 1: **Overview of DRC.** (a) The model generates foreground and background regions using sampled latent variables $\mathbf{z} = \{\mathbf{z}_1, \mathbf{z}_2\}$. $p_{\beta_k}$, $k = 1, 2$ represents the generator for each region. Of note, the pixel re-assignment function is absorbed in the background generator; see Section 3.2 for details. (b) DRC samples the latent variables $\mathbf{z}$ in an iterative manner. Let $\mathbf{x}$ denote the observed image; we use $\hat{\mathbf{x}}_t$, $t = 0, 1, ...$ to represent the image generated by $p_\beta(\mathbf{x}|\mathbf{z})$ at the $t$-th sampling step. DRC has a two-step workflow for learning unsupervised foreground extractors that resembles the E- and M-step in the classic EM algorithm. In the E-step, it employs gradient-based MCMC sampling to infer the latent variables $\mathbf{z}$ as shown in (b). Of note, only the latent variables $\mathbf{z}$ are updated in this step. In the M-step, the sampled latent variables $\mathbf{z}$ are fed into the model for image generation as shown in (a), where the generators are updated to minimize the reconstruction error.

Generating $\mathbf{x}_i$ based on $\mathbf{w}_i$'s distribution involves two steps: (1) sample $\mathbf{w}_i$ from the distribution $p_\beta(\mathbf{w}_i|\mathbf{z})$, and (2) choose either the foreground model (*i.e.*, $p_{\beta_1}(\mathbf{x}_i|\mathbf{z}_1)$) or the background model (*i.e.*, $p_{\beta_2}(\mathbf{x}_i|\mathbf{z}_2)$) to generate $\mathbf{x}_i$ based on the sampled $\mathbf{w}_i$. As such, this distribution of $\mathbf{x}_i$ is a MoE,

$$p_\beta(\mathbf{x}_i|\mathbf{z}) = \sum_{k=1}^{2} p_\beta(w_{ik}=1|\mathbf{z})p_\beta(\mathbf{x}_i|w_{ik}=1, \mathbf{z}_k) = \sum_{k=1}^{2} \pi_{ik} p_{\beta_k}(\mathbf{x}_i|\mathbf{z}_k), \quad (6)$$

wherein the posterior responsibility of $w_{ik}$ is

$$\gamma_{ik} = p(w_{ik}=1|\mathbf{x}_i, \mathbf{z}) = \frac{\pi_{ik} p_{\beta_k}(\mathbf{x}_i|\mathbf{z}_k)}{\sum_{m=1}^{2} \pi_{im} p_{\beta_m}(\mathbf{x}_i|\mathbf{z}_m)}, \quad k = 1, 2. \quad (7)$$

Using a fully-factorized joint distribution of $\mathbf{x}$, we have $p_\beta(\mathbf{x}|\mathbf{z}) = \prod_{i=1}^{D} \sum_{k=1}^{2} \pi_{ik} p_{\beta_k}(\mathbf{x}_i|\mathbf{z}_k)$ as the generative modeling of $\mathbf{x} \in \mathbb{R}^D$.

**Learning Pixel Re-assignment for Background Modeling** We use pixel re-assignment in the background generative model as the essential inductive bias for modeling the background region. This is partially inspired by the concepts of "texture" and "texton" by Julez [45, 67], where the textural part of an image may contain fewer structural elements in preattentive vision, which coincides with our intuitive observation of the background regions.

We use a separate pair of energy-based prior model $\alpha_{\texttt{pix}}$ and generative model $\beta_{\texttt{pix}}$ to learn the re-assignment. For simplicity, we absorb $\alpha_{\texttt{pix}}$ and $\beta_{\texttt{pix}}$ in the models for background modeling, *i.e.*, $\alpha_2$ and $\beta_2$, respectively. In practice, the re-assignment follows the output of $\beta_{\texttt{pix}}$, a shuffling grid with the same size of the image $\mathbf{x}$. Its values indicate the re-assigned pixel coordinates; see Fig. 2. We find that shuffling the background pixels using the learned re-assignment facilitates the model to capture the regularities of the background regions. Specifically, the proposed model with this essential in-

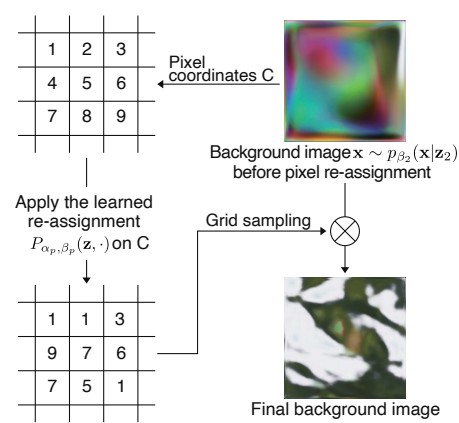

Figure 2: **Pixel re-assignment.** The output of $\beta_p$ can be viewed as a learned re-assignment of the original background pixels that follows the mapped grid $P_{\alpha_p, \beta_p}(\mathbf{z}, C)$. Note that the re-assignment function $P_{\alpha_p, \beta_p}(\mathbf{z}, \cdot)$ might not be injective. The final background image is generated via grid sampling.

ductive bias learns to constantly give the correct mask assignment, whereas most previous fully-unsupervised methods do not provide explicit identification of the foreground and background regions; see discussion in Section 4.1 for more details.

### 3.3 Deep Region Competition: from Generative Modeling to Foreground Extraction

The complete-data distribution from the image modeling is

$$
\begin{aligned}
p_\theta(\mathbf{x}, \mathbf{z}, \mathbf{w}) &= p_\beta(\mathbf{x}|\mathbf{w}, \mathbf{z}) p_\beta(\mathbf{w}|\mathbf{z}) p_\alpha(\mathbf{z}) \\
&= \left( \prod_{i=1}^{D} \prod_{k=1}^{2} p_{\beta_k}(\mathbf{x}_i|\mathbf{z}_k)^{w_{ik}} \right) \left( \prod_{i=1}^{D} \prod_{k=1}^{2} \pi_{ik}^{w_{ik}} \right) p_\alpha(\mathbf{z}) \\
&= p_\alpha(\mathbf{z}) \prod_{i=1}^{D} \prod_{k=1}^{2} \left( \pi_{ik} p_{\beta_k}(\mathbf{x}_i|\mathbf{z}_k) \right)^{w_{ik}},
\end{aligned}
\tag{8}
$$

where $p_\alpha(\mathbf{z}) = p_{\alpha_1}(\mathbf{z}_1) p_{\alpha_2}(\mathbf{z}_2)$ is the prior model given by LEBMs. $\alpha = \{\alpha_1, \alpha_2\}$, and $\theta = \{\alpha, \beta\}$. $\mathbf{w}$ is the vector of $(\mathbf{w}_i)$, $i = 1, ...D$, whose joint distribution is assumed to be fully-factorized.

Next, we derive the complete-data log-likelihood as our learning objective:

$$
\mathcal{L}(\theta) = \log p_\theta(\mathbf{x}, \mathbf{z}, \mathbf{w}) = \log p_\alpha(\mathbf{z}) + \sum_{i=1}^{D} \sum_{k=1}^{2} w_{ik} \left( \log \pi_{ik} + \log p_{\beta_k}(\mathbf{x}_i|\mathbf{z}_k) \right).
\tag{9}
$$

Of note, $\mathbf{w}$ and $\mathbf{z}$ are unobserved variables in the modeling, which makes it impossible to learn the model directly through MLE. To calculate the gradients of $\theta$, we instead optimize $\mathbf{E}_{\mathbf{z} \sim p(\mathbf{z}|\mathbf{x}), \mathbf{w} \sim p(\mathbf{w}|\mathbf{x}, \mathbf{z})}[\mathcal{L}(\theta)]$ based on the fact that underlies the EM algorithm:

$$
\begin{aligned}
\nabla_\theta \log p_\theta(\mathbf{x}) &= \int_{\mathbf{z}} p_\theta(\mathbf{z}|\mathbf{x}) d\mathbf{z} \int_{\mathbf{w}} p_\theta(\mathbf{w}|\mathbf{z}, \mathbf{x}) \nabla_\theta \log p_\theta(\mathbf{x}, \mathbf{z}, \mathbf{w}) d\mathbf{w} \\
&= \mathbf{E}_{\mathbf{z} \sim p_\theta(\mathbf{z}|\mathbf{x}), \mathbf{w} \sim p_\theta(\mathbf{w}|\mathbf{x}, \mathbf{z})}[\nabla_\theta \log p_\theta(\mathbf{x}, \mathbf{z}, \mathbf{w})].
\end{aligned}
\tag{10}
$$

Therefore, the derived surrogate learning objective becomes

$$
\max_\theta \mathbf{E}_{\mathbf{z} \sim p_\theta(\mathbf{z}|\mathbf{x})} \left[ \mathcal{J}(\theta) \right], \text{ s.t. } \forall i, \sum_{k=1}^{2} \pi_{ik} = 1,
\tag{11}
$$

$$
\mathcal{J}(\theta) = \underbrace{\log p_\alpha(\mathbf{z})}_{\text{objective for LEBM}} + \underbrace{\sum_{i=1}^{D} \sum_{k=1}^{2} \gamma_{ik} \log \pi_{ik}}_{\text{foreground-background partitioning}} + \underbrace{\sum_{i=1}^{D} \sum_{k=1}^{2} \gamma_{ik} \log p_{\theta_k}(\mathbf{x}_i|\mathbf{z}_k)}_{\text{objective for image generation}},
\tag{12}
$$

where $\mathcal{J}(\theta) = \mathbf{E}_{\mathbf{w} \sim p_\theta(\mathbf{w}|\mathbf{x}, \mathbf{z})}[\mathcal{L}(\theta)]$ is the conditional expectation of $\mathbf{w}$, which can be calculated in closed form; see the supplementary material for additional details.

Eq. (11) has an intuitive interpretation. We can decompose the learning objective into three components as in Eq. (12). In particular, the second term $\sum_{i=1}^{D} \sum_{k=1}^{2} \gamma_{ik} \log \pi_{ik}$ has a similar form to the cross-entropy loss commonly used for supervised segmentation task, where the posterior responsibility $\gamma_{ik}$ serves as the target distribution. It is as if the foreground and background generative models compete with each other to fit the distribution of each pixel $\mathbf{x}_i$. If the pixel value at $\mathbf{x}_i$ fits better to the distribution of foreground, $p_{\beta_1}(\mathbf{x}_i|\mathbf{z}_1)$, than to that of background, $p_{\beta_2}(\mathbf{x}_i|\mathbf{z}_2)$, the model tends to assign that pixel to the foreground region (see Eq. (7)), and vice versa. This mechanism is similar to the process derived in Zhu and Yuille [1], which is the reason why we coin our method Deep Region Competition (DRC).

Prior to our proposal, several methods [1, 37, 38] also employ mixture models and competition among the components to perform unsupervised foreground or image segmentation. The original Region Competition [1] combines several families of image modeling with Bayesian inference but is limited by the expressiveness of the pre-specified probability distributions. More recent methods, including IODINE [37] and Slot-attention [38], learn amortized inference networks for latent variables and induce the independence of foreground and background representations using an identical generator. Our method combines the best of the two worlds, reconciling the expressiveness of learned generators with the regularity of generic texture modeling under the framework of LEBM.

To optimize the learning objective in Eq. (11), we approximate the expectation by sampling from the prior $p_\alpha(\mathbf{z})$ and posterior model $p_\theta(\mathbf{z}|\mathbf{x}) \propto p_\alpha(\mathbf{z})p_\beta(\mathbf{x}|\mathbf{z})$, followed by calculating the Monte Carlo average. We use Langevin dynamics [64] to draw persistent MCMC samples, which iterates

$$\mathbf{z}_{t+1} = \mathbf{z}_t + s\nabla_\mathbf{z} \log Q(\mathbf{z}_t) + \sqrt{2s}\epsilon_t, \tag{13}$$

where $t$ is the Langevin dynamics's time step, $s$ the step size, and $\epsilon_t$ the Gaussian noise. $Q(\mathbf{z})$ is the target distribution, being either $p_\alpha(\mathbf{z})$ or $p_\theta(\mathbf{z}|\mathbf{x})$. $\nabla_\mathbf{z} \log Q(\mathbf{z}_t)$ is efficiently computed via automatic differentiation in modern learning libraries [68]. We summarize the above process in Algorithm 1.

---

**Algorithm 1: Learning models of DRC via EM.**

---

**Input:** Learning iterations $T$, initial parameters for LEBMs $\alpha^{(0)} = \{\alpha_1^{(0)}, \alpha_2^{(0)}\}$ and generators
$\beta^{(0)} = \{\beta_1^{(0)}, \beta_2^{(0)}\}$, $\theta^{(0)} = \{\alpha^{(0)}, \beta^{(0)}\}$, learning rate $\eta_\alpha$ for LEBMs, $\eta_\beta$ for foreground and background generators, observed examples $\{\mathbf{x}^{(i)}\}_{i=1}^N$, batch size $M$, and initial latent variables $\{\mathbf{z}_-^{(i)} = \{\mathbf{z}_{1-}^{(i)}, \mathbf{z}_{2-}^{(i)}\} \sim p_0(\mathbf{z})\}_{i=1}^N$ and $\{\mathbf{z}_+^{(i)} = \{\mathbf{z}_{1+}^{(i)}, \mathbf{z}_{2+}^{(i)}\} \sim p_0(\mathbf{z})\}_{i=1}^N$.

**Output:** $\theta^{(T)} = \{\alpha_1^{(T)}, \beta_1^{(T)}, \alpha_2^{(T)}, \beta_2^{(T)}\}$.

1 **for** $t = 0 : T - 1$ **do**

2     Sample a minibatch of data $\{\mathbf{x}^{(i)}\}_{i=1}^M$;

3     **Prior sampling for learning LEBMs:** For each $\mathbf{x}^{(i)}$, update $\mathbf{z}_-^{(i)}$ using Eq. (13), with target distribution $\pi(\mathbf{z}) = p_{\alpha^{(t)}}(\mathbf{z})$;

4     **Posterior sampling for foreground and background generation:** For each $\mathbf{x}^{(i)}$, update $\mathbf{z}_+^{(i)}$ using Eq. (13), with target distribution $Q(\mathbf{z}) = p_{\theta^{(t)}}(\mathbf{z}|\mathbf{x})$;

5     **Update LEBMs:** $\alpha^{(t+1)} = \alpha^{(t)} + \eta_\alpha \frac{1}{m} \sum_{i=1}^m [\nabla_\alpha f_{\alpha^{(t)}}(\mathbf{z}_+^{(i)}) - \nabla_\alpha f_{\alpha^{(t)}}(\mathbf{z}_-^{(i)})]$;

6     **Update foreground and background generators:**
    $\beta^{(t+1)} = \beta^{(t)} + \eta_\beta \frac{1}{m} \sum_{i=1}^m \nabla_\beta \log p_{\beta^{(t)}}(\mathbf{x}^{(i)}|\mathbf{z}_+^{(i)})$;

---

During inference, we initialize the latent variables $\mathbf{z}$ for MCMC sampling from Gaussian white noise and run only the posterior sampling step to obtain $\mathbf{z}_+$. The inferred mask and region images are then given by the outputs of generative models $p_{\beta_k}(\mathbf{w}|\mathbf{z}_+)$ and $p_{\beta_k}(\mathbf{x}|\mathbf{z}_+)$, $k = 1, 2$, respectively.

### 3.4 Technical Details

**Pseudo label for additional regularization**    Although the proposed DRC explicitly models the interaction between the regions, it is still possible that the model converges to a trivial extractor, which treats the entire image as the foreground or background region, leaving the other region null. We exploit the symbolic vector $\mathbf{y}$ emitted by the LEBM (see Section 3.1) for additional regularization. The strategy is similar to the mutual information maximization used in InfoGAN [69]. Specifically, we use the symbolic vector $\mathbf{y}$ inferred from $\mathbf{z}$ as the pseudo-class label for $\mathbf{z}$ and train an auxiliary classifier jointly with the above models; it ensures that the generated regions $\mathbf{x}_k$ contain similar symbolic information for $\mathbf{z}_k$. Intuitively, this loss prevents the regions from converging to null since the symbolic representation $\mathbf{y}_k$ would never be well retrieved if that did happen.

**Implementation**    We adopt a similar architecture for the generator as in DCGAN [70] throughout the experiments and only change the dimension of the latent variables $\mathbf{z}$ for different datasets. The generator consists of a fully connected layer followed by five stacked upsample-conv-norm layers. We replace the batch-norm layers [71] with instance-norm [72] in the architecture. The energy-term in LEBM is parameterized by a 3-layered MLP. We adopt orthogonal initialization [73] commonly used in generative models to initialize the networks and orthogonal regularization [74] to facilitate training. In addition, we observe performance improvement when adding Total-Variation norm [75] for the background generative model. More details, along with specifics of the implementation used in our experiments, are provided in the supplementary material.

## 4   Experiments

We design experiments to answer three questions: (1) How does the proposed method compare to previous state-of-the-art competitors? (2) How do the proposed components contribute to the model performance? (3) Does the proposed method exhibit generalization on images containing unseen instances (*i.e.*, same category but not the same instance) and even objects from novel categories?

To answer these questions, we evaluate our method on five challenging datasets in two groups: (1) Caltech-UCSD Birds-200-2011 (Birds) [76], Stanford Dogs (Dogs) [77], and Stanford Cars (Cars) [78] datasets; (2) CLEVR6 [79] and Textured Multi-dSprites (TM-dSprites) [80] datasets. The first group of datasets covers complex real-world domains, whereas the second group features environments of the multi-object foreground with challenging spatial configurations or confounding backgrounds. As to be shown, the proposed method is generic to various kinds of input and produces more competitive foreground-background partition results than prior methods.

## 4.1    Results on Foreground Extraction

| Model | Single Object | | | | | | Multi-Object | | | |
| | Birds | | Dogs | | Cars | | CLEVR6 | | TM-dSprites | |
| | IoU | Dice | IoU | Dice | IoU | Dice | IoU | Dice | IoU | Dice |
|---|---|---|---|---|---|---|---|---|---|---|
| W-Net* | 24.8 | 38.9 | 47.7 | 62.1 | 52.8 | 67.6 | - | - | - | - |
| GrabCut | 30.2 | 42.7 | 58.3 | 70.9 | 61.3 | 73.1 | 19.0 | 30.5 | 61.9 | 71.0 |
| ReDO§ | 46.5 | 60.2 | 55.7 | 70.3 | 52.5 | 68.6 | 18.6 | 31.0 | 9.4 | 17.2 |
| OneGAN*† | 55.5 | 69.2 | 71.0 | 81.7 | 71.2 | 82.6 | - | - | - | - |
| IODINE§ | 30.9 | 44.6 | 54.4 | 67.0 | 51.7 | 67.3 | 19.9 | 32.4 | 7.3 | 12.8 |
| Slot-Attn.§ | 35.6 | 51.5 | 38.6 | 55.3 | 41.3 | 58.3 | 83.6 | 90.7 | 7.3 | 13.5 |
| Ours | **56.4** | **70.9** | **71.7** | **83.2** | **72.4** | **83.7** | **84.7** | **91.5** | **78.8** | **87.5** |

Table 1: **Foreground extraction results on training data measured in IoU and Dice.** Higher is better in all scores. *Results of W-Net and OneGAN are provided by Benny and Wolf [31]. Of note, results of these two models on Dogs and Cars datasets may **not** be directly comparable to other listed methods, as the data used for training and evaluation could be different. We include these results as a rough reference since no official implementation or pretrained model are publicly available. § indicates unfair baseline results obtained using extra ground-truth information, *i.e.*, we choose the best-matching scores from the permutation of foreground and background masks. †OneGAN is a strong **weakly supervised** baseline, which requires clean background images to provide additional supervision. We include this model as a potential upper bound of the fully unsupervised methods.

**Single object in the wild**    In the first group of datasets, there is typically a single object in the foreground, varying in shapes, texture, and lighting conditions. Unsupervised foreground extraction on these datasets requires much more sophisticated visual cues than colors and shapes. Birds dataset consists of 11,788 images of 200 classes of birds annotated with high-quality segmentation masks, Dogs dataset consists of 20,580 images of 120 classes annotated with bounding boxes, and Cars dataset consists of 16,185 images of 196 classes. The latter two datasets are primarily made for fine-grained categorization. To evaluate foreground extraction, we follow the practice in Benny and Wolf [31], and approximate ground-truth masks for the images with Mask R-CNN [16], pre-trained on the MS COCO [9] dataset with a ResNet-101 [81] backend. The pre-trained model is acquired from the detectron2 [82] toolkit. This results in 5,024 dog images and 12,322 car images with a clear foreground-background setup and corresponding masks.

On datasets featuring a single foreground object, we use the 2-slot version of IODINE and Slot-attention. Since ReDO, IODINE, and Slot-Attention do not distinguish foreground and background in output regions, we choose the best-matching scores from the permutation of foreground and background masks as in [28]. We observe that the proposed method and Grabcut are the only two methods that provide explicit identification of foreground objects and background regions. While the Grab-cut algorithm actually requires a predefined bounding box as input that specifies the foreground region, our method, thanks to the learned pixel re-assignment (see Section 3.2), can achieve this in a fully unsupervised manner. Results in Table 1 show that our method outperforms all the unsupervised baselines by a large margin, exhibiting comparable performance even to the weakly supervised baseline that requires additional background information as inputs [31]. We provide samples of foreground extraction results as well as generated background and foreground regions in Fig. 3. Note that our final goal is not to synthesize appealing images but to learn foreground extractors in a fully unsupervised manner. As the limitation of our method, DRC generates foreground and background regions less realistic than those generated by state-of-the-art GANs, which hints a possible direction for future work. More detailed discussions of the limitation can be found in supplementary material.

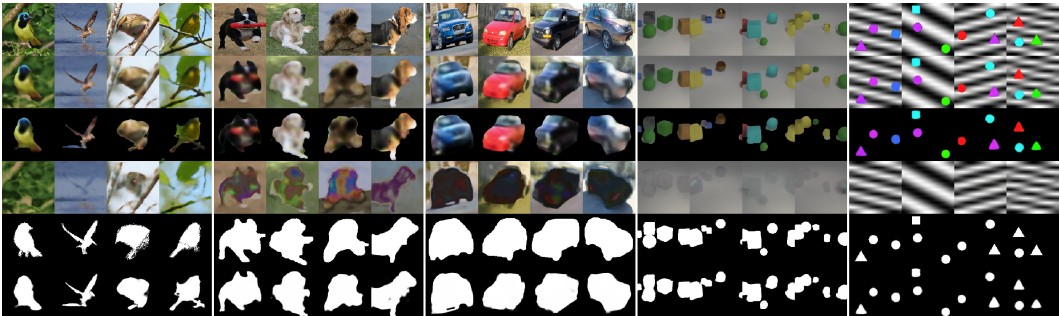

Figure 3: **Foreground extraction results for each dataset**; zoom in for better visibility. From top to bottom: (i) observed images, (ii) generated images, (iii) masked generated foregrounds, (iv) generated backgrounds, (v) ground-truth foreground masks, and (vi) inferred foreground masks. More samples and results of baselines can be found in the supplementary material.

**Multi-object scenes** The second group of datasets contains images with possibly simpler foreground objects but more challenging scene configurations or background parts. Visual scenes in the CLEVR6 dataset contain various objects and often with partial occlusions and truncations. Following the evaluation protocol in IODINE and Slot-attention, we use the first 70K samples from CLEVR [79] and filter the samples for scenes with at most 6 objects for training and evaluation, *i.e.*, CLEVR6. The TM-dSprites dataset is a variant of Multi-dSprites [80] but has strongly confounding backgrounds borrowed from Textured MNIST [32]. We generate 20K samples for the experiments. Similar to Greff et al. [37] and Locatello et al. [38], we evaluate on a subset containing 1K samples for testing. Note that IODINE and Slot-attention are designed for segmenting complex multi-object scenes using slot-based object representations. Ideally, the output of these models consists of masks for each individual object, while the background is viewed as a virtual "object" as well. In practice, however, it is possible that the model distributes the background over all the slots as mentioned in Locatello et al. [38]. We therefore propose two corresponding approaches (see the supplementary material for more details) to convert the output object masks into a foreground-background partition and report the best results of these two options for IODINE and Slot-attention in Table 1.

On the CLEVR6 dataset, we use the publicly available pretrained model for IODINE, which achieves a reasonable ARI (excluding background pixels) of 94.4 on the testing data, close to the testing results in Greff et al. [37]. We observe that IODINE distributes the background over all the slots for some of the testing samples, resulting in much lower IoU and Dice scores. We re-train the Slot-attention model using the official implementation on CLEVR6, as no pretrained model is publicly available. The re-trained model achieves a foreground ARI of 98.0 on the 1K testing samples, which we consider as a sign of valid re-implementation. Results in Table 1 demonstrate that the proposed method can effectively process images of challenging multi-object scenes. To be specific, our method demonstrates competitive performance on the CLEVR6 dataset compared with the SOTA object discovery method. Moreover, as shown empirically in Fig. 3, the proposed method can handle the strongly confounding background introduced in Greff et al. [32], whereas previous methods are distracted by the background and mostly fail to capture the foreground objects.

### 4.2 Ablation Study

We provide ablation studies on the Birds dataset to inspect the contribution of each proposed component in our model. As shown in Table 2, we observe that replacing the LEBMs in the foreground and background models with amortized inference networks significantly harms the performance of the proposed method. In particular, the modified model fails to generate any meaningful results (marked as - in Table 2). We conjecture that LEBM benefits from the low-dimensionality of the latent space [47] and therefore enjoys more efficient learning. However, the inference networks need to learn an extra mapping from the high-dimensional image space to the latent space and require more elaborate architecture and tuning for convergence. Furthermore, we observe that the

| Model | IoU | Dice |
|---|---|---|
| amortized inference* | - | - |
| w/o pix. re-assign. | 21.8 | 35.3 |
| w/o pseudo label | 48.7 | 64.2 |
| w/o TV-norm reg. | 53.0 | 68.1 |
| w/o ortho. reg. | 54.3 | 69.2 |
| short-run chain† | 52.5 | 67.7 |
| Full model | 56.4 | 70.9 |

Table 2: **Ablation study on Birds.** *We replace the LEBM with encoders to perform amortized inference for the latent variables **z** within a variational framework as in VAE [83]. †We explore the possibility of using short-run MCMC [84] instead of persistent chain sampling.

model that does not learn pixel re-assignment for background can still generate meaningful images but only vaguely captures masks for foreground extraction.

## 4.3 Generalizable Foreground Extraction

**Extracting novel foreground objects from training categories**    We show results on generalizing to novel objects from the training classes. To evaluate our method, we split the Birds dataset following Chen et al. [28], resulting in 10K training images and 1K testing images. On Dogs and Cars datasets, we split the dataset based on the original train-test split [77, 78]. This split gives 3,286 dog images and 6,218 car images for training, and 1,738 dog images and 6,104 car images for testing, respectively. As summarized in Table 3, our method shows superior performances compared with baselines; the performance gap between training and testing is constantly small over all datasets.

| Model | Birds | | Dogs | | Cars | |
|---|---|---|---|---|---|---|
| | IoU | Dice | IoU | Dice | IoU | Dice |
| | Tr.\|Te. | Tr.\|Te. | Tr.\|Te. | Tr.\|Te. | Tr.\|Te. | Tr.\|Te. |
| GrabCut* | 30.2\|30.3 | 42.7\|42.8 | 58.3\|57.9 | 70.8\|70.5 | 60.9\|61.6 | 72.7\|73.5 |
| ReDO | 46.8\|47.1 | 61.4\|61.7 | 54.3\|52.8 | 69.2\|67.9 | 52.6\|52.5 | 68.7\|68.6 |
| Ours | **54.8\|54.6** | **69.5\|69.4** | **71.6\|72.3** | **83.2\|83.6** | **71.9\|70.8** | **83.3\|82.5** |

Table 3: **Performance of DRC on training and held-out testing data.** *Note that GrabCut is a deterministic method that does not require training. We report the results of GrabCut [5] on these splits only for reference. Tr. indicates the performance on training data, and Te. indicates the performance on testing data.

**Extracting novel foreground objects from unseen categories**    To investigate how well our method generalizes to categories unseen during training, we evaluate the models trained in real-world single object datasets on the held-out testing data from different categories. We use the same training and testing splits on these datasets as in the previous experiments. Table 4 shows that our method outperforms the baselines on the Birds dataset when the model has trained on Dogs or Cars dataset, which have quite different distributions in foreground object shapes. Competitors like ReDO also exhibit such out-of-distribution generalization but only to a limited extent. Similar results are observed when using Dogs or Cars as the testing set. We can see that when the model is trained on Dogs and evaluated on Cars or vice versa, it still maintains comparable performances w.r.t. those are trained&tested on the same class. We hypothesize that these two datasets have similar distributions in fore-

| Test | Train | GrabCut | | ReDO | | Ours | |
|---|---|---|---|---|---|---|---|
| | | IoU | Dice | IoU | Dice | IoU | Dice |
| | Birds* | | | 47.1 | 61.7 | 54.6 | 69.4 |
| Birds | Dogs | 30.3 | 42.8 | 22.2 | 35.3 | **41.3** | **57.4** |
| | Cars | | | 16.4 | 27.7 | 39.2 | 55.3 |
| | Dogs* | | | 52.8 | 67.9 | 72.3 | 83.6 |
| Dogs | Cars | 57.9 | 70.5 | 44.5 | 61.2 | **67.8** | **80.4** |
| | Birds | | | 44.0 | 60.3 | 53.6 | 69.1 |
| | Cars* | | | 52.5 | 68.6 | 70.8 | 82.5 |
| Cars | Dogs | 61.6 | 73.5 | 51.6 | 67.1 | **68.6** | **81.0** |
| | Birds | | | 41.8 | 58.6 | 45.1 | 61.7 |

Table 4: **Performance of DRC on unseen testing categories.** *We include the testing results of models trained with data from the same categories for reference.

ground objects and background regions. In the light of this, we can further entail that the distribution of Dogs is most similar to that of Cars and less similar to that of Birds according to the results, which is consistent with our intuitive observation of the data. We provide a preliminary analysis of the statistics of these datasets in the supplementary material.

## 5 Conclusion

We have presented the Deep Region Competition, an EM-based fully unsupervised foreground extraction algorithm fueled by energy-based prior and generative image modeling. We propose learned pixel re-assignment as an inductive bias to capture the background regularities. Experiments demonstrate that DRC exhibits more competitive performances on complex real-world data and challenging multi-object scenes. We show empirically that learned pixel re-assignment helps to provide explicit identification for foreground and background regions. Moreover, we find that DRC can potentially generalize to novel foreground objects even from categories unseen during training. We hope our work will inspire future research along this challenging but rewarding research direction.

## Acknowledgements

The work was supported by NSF DMS-2015577, ONR MURI project N00014-16-1-2007, and DARPA XAI project N66001-17-2-4029. We would like to thank Bo Pang from the UCLA Statistics Department for his insights on the latent-space energy-based model and four anonymous reviewers (especially Reviewer RcgA) for their constructive comments.

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
