# Supplementary of Unsupervised Foreground Extraction via Deep Region Competition

## Contents

# A    Dataset Details

## A.1    Caltech-UCSD Birds-200-2011

Caltech-UCSD Birds-200-2011 (Birds) dataset consists of 11,788 images of 200 classes of birds annotated with high-quality segmentation masks. Each image is further annotated with 15 part locations, 312 binary attributes, and 1 bounding box. We use the provided bounding box to extract a center square from the image, and scale it to $128 \times 128$ pixels. Each scene contains exactly one foreground object.

## A.2    Stanford Dogs

Stanford Dogs (Dogs) dataset consists of 20,580 images of 120 classes annotated with bounding boxes. We first use the provided bounding box to extract the center square, and then scale it to $128 \times 128$ pixels. As stated in the paper, we approximate ground-truth masks for the pre-processed images with Mask R-CNN [1], pre-trained on the MS COCO [2] dataset with a ResNet-101 [3] backend. The pretrained model is acquired from the detectron2 [4] toolkit. We exclude the images where no dog is detected. We then manually exclude those images where the foreground object has occupied more than $\sim 90\%$ of the image, those with poor masks, and those with significant foreground distractors such as humans (see Fig. 1). The filtering strategy results in 5,024 images with a clear foreground-background setup and high-quality mask.

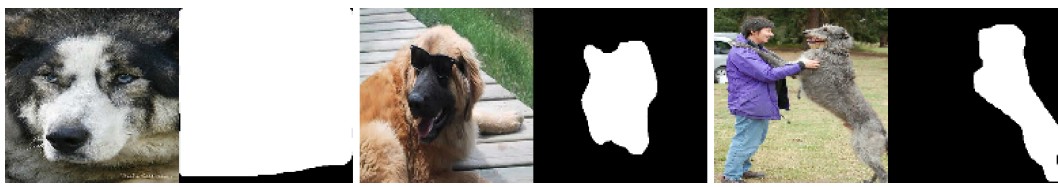

Figure 1: Examples of excluded images. From left to right: (i) image with a foreground object that occupied too much space, (ii) image with a low-quality mask, and (iii) image with significant foreground distractors.

## A.3    Stanford Cars

Stanford Cars (Cars) dataset consists of 16,185 images of 196 classes annotated with bounding boxes. Though also being primarily designed for fine-grained categorization, it has a much clearer foreground-background setup compared with the Dogs dataset. We employ a similar process as used for Dogs dataset to approximate the ground-truth masks, and only exclude those images where cars are not properly detected. It finally produces 12,322 images for our experiments.

## A.4    CLEVR6

CLEVR6 dataset is a subset of the original CLEVR dataset [5] with masks, generated by Greff et al. [6]. We follow the evaluation protocol adopted by IODINE [6] and Slot-attention [7], which takes the first 70K samples from CLEVR. These samples are then filtered to only include scenes with at most 6 objects. Additionally, we perform a center square crop of $192 \times 192$ from the original $240 \times 320$ image, and scale it to $128 \times 128$ pixels. The resulting CLEVR6 dataset contains 3-6 foreground objects that could be with partial occlusion and truncation in each visual scene.

## A.5    Textured Multi-dSprites

Textured Multi-dSprites (TM-dSprites) dataset, which is based on the dSprites dataset [8] and Textured MNIST [9], consists of 20,000 images with a resolution of $128 \times 128$. Each image contains 2-3 random sprites, which vary in terms of shape (square, circle, or triangle), color (uniform saturated colors), and position (continuous). The background regions are borrowed from Textured MNIST dataset [9]. The textures for the background are randomly shifted samples from a bank of 20 sinusoidal textures with different frequencies and orientations. We adopt a simpler foreground setting compared with the vanilla Multi-dSprites dataset used in [6], *i.e.*, the foreground objects are not occluded as the dataset is designed to emphasize the background part. Some samples are presented in Fig. 2.

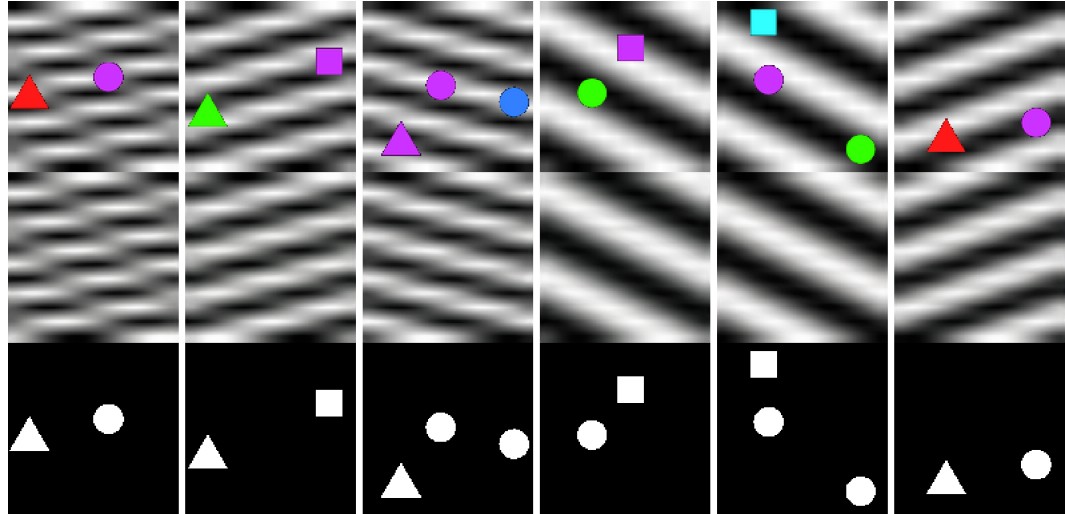

Figure 2: Samples from TM-dSprites. From top to bottom: (1) observed images, (ii) background textures, and (iii) ground-truth masks.

## B  Details on Models and Hyperparameters

**Architecture**  As mentioned in the paper, we use the same overall architecture for different datasets (while the size of latent variables may vary). The details for the generators and LEBMs are summarized in the Table 1 and Table 2.

| Dataset | Foreground | Background | Pixel Re-assignment |
|---|---|---|---|
| Birds | 256 | 256 | 512 |
| Dogs | 256 | 256 | 512 |
| Cars | 256 | 192 | 512 |
| CLEVR6 | 256 | 2 | 256 |
| TM-dSprites | 256 | 4 | 1024 |

Table 1: Dimension of latent variables on each dataset.

**Hyperparameters and Training Details**  For the Langevin dynamics sampling [10], we use $K_0$ and $K_1$ to denote the number of prior and posterior sampling steps with step sizes $s_0$ and $s_1$ respectively. Our hyperparameter choices are: $K_0 = 60, K_1 = 40, s_0 = 0.4$ and $s_1 = 0.1$. These are identical across different datasets. During testing, we set the posterior sampling steps to 300 for Dogs and Cars, and 2.5K, 5K and 5K for Birds, CLEVR6 and TM-dSprites respectively. The parameters of the generators and LEBMs are initialized with orthogonal initialization [11]. The gain is set to 1.0 for all the models. We use the ADAM optimizer [12] with $\beta_1 = 0.5$ and $\beta_2 = 0.999$. Generators are trained with a constant learning rate of 0.0001, and LEBMs with 0.00002. We run experiments on a single V100 GPU with 16GB of RAM and with a batch size of 48. We set the maximum training iterations to 10K and run for at most 48hrs for each dataset.

| Layers | In-Out size | Comment |
|---|---|---|
| *LEBM for Foreground/Background Models* | | |
| Input: $\mathbf{z}$ | $D*$ | |
| Linear, LReLU | 200 | |
| Linear, LReLU | 200 | |
| Linear | $K^\dagger$ | |
| *LEBM for Pixel Re-assignment Model* | | |
| Input: $\mathbf{z}$ | $D*$ | |
| Linear, LReLU | 200 | |
| Linear, LReLU | 200 | |
| Linear, LReLU | 200 | |
| Linear | 1 | |
| *Generator for Foreground/Background Model and Re-assignment Model* | | |
| Input: $\mathbf{z}$ | $D*$ | |
| Linear, LReLU | $4 \times 4 \times 128$ | reshaped output |
| UpConv3x3Norm, LReLU | $8 \times 8 \times 1024$ | stride 1 & padding 1 |
| UpConv3x3Norm, LReLU | $16 \times 16 \times 512$ | stride 1 & padding 1 |
| UpConv3x3Norm, LReLU | $32 \times 32 \times 256$ | stride 1 & padding 1 |
| UpConv3x3Norm, LReLU | $64 \times 64 \times 128$ | stride 1 & padding 1 |
| UpConv3x3Norm, LReLU | $128 \times 128 \times 64$ | stride 1 & padding 1 |
| Conv3x3 | $128 \times 128 \times (3+1)$ | RGB & Mask |
| | $128 \times 128 \times 2$ | Re-assignment grid |
| *Auxiliary classifier for Foreground/Background Model* | | |
| Input: $\mathbf{x}$ | $128 \times 128 \times 3$ | generated image |
| Conv4x4Norm, LReLU | $64 \times 64 \times 64$ | stride 2 & padding 1 |
| Conv4x4Norm, LReLU | $32 \times 32 \times 128$ | stride 2 & padding 1 |
| Conv4x4Norm, LReLU | $16 \times 16 \times 256$ | stride 2 & padding 1 |
| Conv4x4Norm, LReLU | $8 \times 8 \times 512$ | stride 2 & padding 1 |
| Conv4x4Norm, LReLU | $4 \times 4 \times 1024$ | stride 2 & padding 1 |
| Conv4x4 | $1 \times 1 \times K^\dagger$ | |

Table 2: Architecture of the generators, LEBMs and auxiliary classifiers (see Appendix C.2). UpConv3x3Norm denotes a Upsampling-Convolutional-InstanceNorm layer with a convolution kernel size of 3. Similarly, Conv4x4Norm denotes a Convolutional-InstanceNorm layer with a kernel size of 4. LReLU denotes the Leaky-ReLU activation function. The leak factor for LReLU is 0.2 in LEBMs and auxiliary classifiers, and 0.01 in generators. *$D$ represents the dimensions of the latent variables for different datasets; see Table 1. †$K$ represents the pre-specified category number for latent variables. We use 200 for both the foreground and background LEBMs on real-world datasets, and 30 and 10 in the foreground and background LEBMs on multi-object datasets respectively.

## C  Details on Learning Objective and Regularization

### C.1  Learning Objective

**Derivation of Surrogate Learning Objective**  $\mathcal{J}(\theta) = \mathbf{E}_{\mathbf{w} \sim p_\beta(\mathbf{w}|\mathbf{x},\mathbf{z})} \left[\mathcal{L}(\theta)\right]$ is the conditional expectation of $\mathbf{w}$,

$$
\begin{aligned}
\mathcal{J}(\theta) &= \mathbf{E}_{\mathbf{w} \sim p_\beta(\mathbf{w}|\mathbf{x},\mathbf{z})} \left[\mathcal{L}(\theta)\right] \\
&= \log p_\alpha(\mathbf{z}) + \mathbf{E}\left[\sum_{i=1}^{D}\sum_{k=1}^{2} w_{ik}\left(\log \pi_{ik} + \log p_{\beta_k}(\mathbf{x}_i|\mathbf{z}_k)\right)\right] \\
&= \log p_\alpha(\mathbf{z}) + \sum_{i=1}^{D}\sum_{k=1}^{2} \mathbf{E}\left[w_{ik}\right]\left(\log \pi_{ik} + \log p_{\beta_k}(\mathbf{x}_i|\mathbf{z}_k)\right),
\end{aligned}
\tag{1}
$$

where $\mathbf{E}$ is the conditional expectation of $\mathbf{w}$. Recall that $w_{ik} \in \{0,1\}$. The expectation becomes

$$
\begin{aligned}
\mathbf{E}\left[w_{ik}\right] &= 0 \times p(w_{ik}=0|\mathbf{x}_i,\mathbf{z}) + 1 \times p(w_{ik}=1|\mathbf{x}_i,\mathbf{z}) \\
&= \gamma_{ik},
\end{aligned}
\tag{2}
$$

which is the posterior responsibility of $w_{ik}$. We can further decompose $\mathcal{J}(\theta)$ into

$$\mathcal{J}(\theta) = \underbrace{\log p_\alpha(\mathbf{z})}_{\text{objective for LEBM}} + \underbrace{\sum_{i=1}^{D}\sum_{k=1}^{2} \gamma_{ik}\log\pi_{ik}}_{\text{foreground-background partitioning}} + \underbrace{\sum_{i=1}^{D}\sum_{k=1}^{2}\gamma_{ik}\log p_{\beta_k}(\mathbf{x}_i|\mathbf{z}_k)}_{\text{objective for image generation}}, \tag{3}$$

as mentioned in the paper.

**Understanding the Optimization Process**    Note that the surrogate learning objective is an expectation w.r.t $\mathbf{z}$,

$$\max_{\theta}\ \mathbf{E}_{\mathbf{z}\sim p_\theta(\mathbf{z}|\mathbf{x})}\left[\mathcal{J}(\theta)\right],\ \text{s.t.}\ \forall i,\ \sum_{k=1}^{2}\pi_{ik}=1, \tag{4}$$

which is generally intractable to calculate. We therefore need to approximate the expectation by sampling from the distributions, and calculating the Monte Carlo average. In practice, this can be done by gradient-based MCMC sampling method, such as Langevin Dynamics [10].

Given $\mathbf{x}$, we have $p_\theta(\mathbf{z}|\mathbf{x})\propto p_\beta(\mathbf{x}|\mathbf{z})p_\alpha(\mathbf{z})$. Note that

$$\begin{aligned}
\nabla_{\mathbf{z}}\log p_\beta(\mathbf{x}|\mathbf{z}) &= \frac{1}{p_\beta(\mathbf{x}|\mathbf{z})}\nabla_{\mathbf{z}}p_\beta(\mathbf{x}|\mathbf{z}) \\
&= \int_{\mathbf{w}} p_\beta(\mathbf{w}|\mathbf{x},\mathbf{z})\nabla_{\mathbf{z}}\log p_\beta(\mathbf{x},\mathbf{w}|\mathbf{z})d\mathbf{w} \\
&= \mathbf{E}_{\mathbf{w}\sim p_\beta(\mathbf{w}|\mathbf{x},\mathbf{z})}\left[\nabla_{\mathbf{z}}\log p_\beta(\mathbf{x},\mathbf{w}|\mathbf{z})\right].
\end{aligned} \tag{5}$$

Therefore, the log-likelihood of surrogate target distribution for the Langevin dynamics at the $t$-th step is

$$\begin{aligned}
\log\tilde{Q}(\mathbf{z}_t) &= \log p_\alpha(\mathbf{z}_t) + \mathbf{E}_{\mathbf{w}\sim p_\beta(\mathbf{w}|\mathbf{x},\mathbf{z}_t)}\left[\sum_{i=1}^{D}\sum_{k=1}^{2} w_{ik}\left(\log\pi_{ik}+\log p_{\beta_k}(\mathbf{x}_i|\mathbf{z}_{k,t})\right)\right] \\
&= \log p_\alpha(\mathbf{z}_t) + \sum_{i=1}^{D}\sum_{k=1}^{2}\gamma_{ik,t}\left(\log\pi_{ik}+\log p_{\beta_k}(\mathbf{x}_i|\mathbf{z}_{k,t})\right),
\end{aligned} \tag{6}$$

which has the same form as $\mathcal{J}(\theta)$. However, instead of updating parameters $\theta$, Langevin dynamics updates the latent variables $\mathbf{z}$ with the calculated gradients.

The two-step learning process of the DRC models can be understood as follows: (1) in the first step, the algorithm optimizes $\mathcal{J}$ by updating latent variables $\mathbf{z}$, where the posterior responsibility $\gamma_{ik}$ inferred at each step serves to gradually disentangle the foreground and background components, and (2) in the second step, the updated $\mathbf{z}$ is fed again into the models to generate the observation $\mathbf{x}$, where the algorithm optimizes $\mathcal{J}$ by updating the model parameters $\theta$.

It is worth mentioning that learning LEBMs requires an extra sampling step [13], as the gradients are given by the following

$$\delta_\alpha(\mathbf{x}) = \mathbf{E}_{p_\theta(\mathbf{z}|\mathbf{x})}\left[\nabla_\alpha f_\alpha(\mathbf{z})\right] - \mathbf{E}_{p_\alpha(\mathbf{z})}\left[\nabla_\alpha f_\alpha(\mathbf{z})\right], \tag{7}$$

where the second terms should be computed by sampling with $p_\alpha(\mathbf{z})$. We term this as *prior sampling* in the main paper.

**Further Details on the Loss Functions**    For the generative models $p_{\beta_k}(\mathbf{x}|\mathbf{z}_k)$, $k=1,2$, we assume that $\mathbf{x} = g_{\beta_k}(\mathbf{z}_k) + \epsilon$, where $g_{\beta_k}(\mathbf{z}_k)$, $k=1,2$ are the generator networks for foreground and background regions, and $\epsilon$ is random noise sampled from a zero-mean Gaussian or Laplace distribution. Assuming a global fixed variance $\sigma^2$ for Gaussian, we have $\log p_{\beta_k}(\mathbf{x}|\mathbf{z}_k) = -\frac{1}{2\sigma^2}\|g_{\beta_k}(\mathbf{z}_k)-\mathbf{x}\|^2 + C$, $k=1,2$, where $C$ is a constant unrelated to $\beta_k$ and $\mathbf{z}_k$. Similarly for Laplace distribution, we have $\log p_{\beta_k}(\mathbf{x}|\mathbf{z}_k) = -\frac{1}{\lambda}|g_{\beta_k}(\mathbf{z}_k)-\mathbf{x}| + C$, $k=1,2$. These two log-likelihoods correspond to the MSE loss and L1 loss commonly used for image reconstruction, respectively.

## C.2 Regularization

**Pseudo Label Learning**  As mentioned in the paper, we exploit the symbolic vector $\mathbf{y}$ emitted by the LEBM for additional regularization. Let the target distribution of $\mathbf{y}_k$ be $P_k$ given by $p_{\alpha_k}(\mathbf{y}|\mathbf{z}_k)$, $k = 1, 2$, which represents the distribution of symbolic vector for foreground and background regions respectively. We can optimize the following objective as a regularization to our original learning objective:

$$\max_{\beta,\tau} \mathcal{L}_{\text{pseudo-label}} = \sum_{k=1}^{2} H(P_k, Q_k), \tag{8}$$

$$H(P_k, Q_k) = -\langle p_{\alpha_k}(\mathbf{y}|\mathbf{z}_k), \log q_{\tau_k}(\mathbf{y}|g_{\beta_k}(\mathbf{z}_k))\rangle, \ k = 1, 2, \tag{9}$$

where $q_{\tau_k}$, $k = 1, 2$ represents the jointly trained auxiliary classifier network (see Appendix B for architecture details) for foreground and background. $g_{\beta_k}(\mathbf{z}_k)$, $k = 1, 2$ represents the output of generator network. We set the weight of this regularization term to $0.1$ for all the models.

**Total Variation norm (TV-norm)**  Total Variation norm [14] is commonly used for image denoising, and has been extended as an effective technique for in-painting. We use TV-norm as a regularization for learning the background generator.

$$\min_{\beta_2} \mathcal{L}_{\text{TV-norm}} = \sum_{h,w} \left( |\frac{\partial g_{\beta_2}(\mathbf{z}_2)}{\partial x}(h, w)| + |\frac{\partial g_{\beta_2}(\mathbf{z}_2)}{\partial y}(h, w)| \right), \tag{10}$$

where $\partial_x g_{\beta_2}(\mathbf{z}_2)(h, w)$ and $\partial_y g_{\beta_2}(\mathbf{z}_2)(h, w)$ represent the horizontal and vertical image gradients at the pixel coordinate $(h, w)$ respectively. We set the weight of this regularization term to $0.01$ for all the models.

**Orthogonal Regularization**  We use orthogonal regularization [15] for the convolutional layers only. Let $\mathbf{W}$ be the flattened kernel weights of the convolutional layers, *i.e.*, the size of $\mathbf{W}$ is $C \times K$ where $C$ is the output channel number. The orthogonal regularization is calculated according to

$$\min_{\beta} \mathcal{L}_{\text{orthogonal-reg}} = \|\mathbf{W}\mathbf{W}^T \odot (\mathbf{1} - \mathbf{I})\|_F, \tag{11}$$

where $\odot$ is the Hadamard product. $\mathbf{I}$ denotes the identity matrix, and $\mathbf{1}$ denotes the matrix filled with ones. We set the weight of this regularization term to $0.1$ for Birds models, and $1.0$ for the rest of the models.

# D   Pytorch-style Code

We provide pytorch-style code to illustrate how the learning and inference in our model work.

**Forward Pass**   In the forward pass, the model takes latent variables **z**, generates foreground and background regions separately, and mixes them into the final image. Note that the pixel re-assignment is applied to both background image and mask. We finds it useful to feed the intermediate feature of background region into the generator for pixel re-assignment.

Listing 1: Forward pass of the DRC model.

```python
def forward(z):
    zf, zb, zs = z[:, :ZF_DIM], \
                 z[:, ZF_DIM:-ZS_DIM] \
                 z[:, -ZS_DIM:]

    ### generating foreground
    fg, fm = fg_net(zf)

    ### generating background
    bg, bm, bg_feat = bg_net(zb)
    shuffling_grid = sp_net(zs, bg_feat.detach())
    bg_shuf = F.grid_sample(bg, shuffling_grid)
    bm_shuf = F.grid_sample(bm, shuffling_grid)

    ### generating foreground masks
    pi = torch.cat([fm_wp, bm_wp], dim=1).softmax(dim=1)
    pi_f, pi_b = pi[:,:1,...], pi[:,1:,...]

    ### mixing regions
    im_p = fg * pi_f + bg_wp * pi_b
    return im_p, fg, bg_shuf, pi_f, pi_b, bg
```

**Sampling Latent Variables**    We employ Langevin Dynamics for sampling latent variables, which iteratively updates the sample with the gradient computed against the likelihood. In the following code, `ebm_net` stands for the LEBMs, which outputs the energy and the distribution paramters of the symbolic vector **y** for the foreground and background regions.

Listing 2: Running Langevin Dynamics for prior and posterior sampling of the latent variables **z**.

```python
def sample_langevin_prior(z):
    ### langevin prior inference
    # only latent variables 'z' are updated
    for __ in range(infer_step_K0):
        z = z.requires_grad_(True)

        en, __, __ = ebm_net(z)
        e_log_lkhd = en.sum() + .5 * z.square().sum()

        d_ebm_z = torch.autograd.grad(e_log_lkhd, z)[0]
        z = z - 0.5 * (self.delta_0 ** 2) * d_ebm_z \
                    + self.delta_0 * torch.randn_like(z)
    return z

def sample_langevin_posterior(z, im_t):
    ### langevin posterior inference
    # only latent variables 'z' are updated
    for __ in range(infer_step):
        z = z.requires_grad_(True)

        im_p, fg, bg_shuf, pi_f, pi_b, bg = forward(z)

        ### log-lkhd for LEBMs
        en, __, __ = ebm_net(z)

        ### log-lkhd for generators
        log_pf = - F.l1_loss(fg, im_t, reduction='none')
                / (2. * SIGMA ** 2)
        log_pb = - F.l1_loss(bg_shuf, im_t, reduction='none')
                / (2. * SIGMA ** 2)

        # posterior responsibility
        with torch.no_grad():
            ga_f = pi_f * log_pf.exp() /
                    (pi_f * log_pf.exp()
                + pi_b * log_pb.exp() + 1e-8)
        # objective for image generation
        e_z_log_p = ga_f.detach() * \
                    ((pi_f + 1e-8).log() + log_pf) \
                + (1. - ga_f.detach()) * \
                    ((pi_b + 1e-8).log() + log_pb)
        # regularization
        tv_norm = tv_loss(bg_shuf)

        j_log_lkhd = - e_z_log_p.sum() + tv_norm * .01 + \
                    + en.sum() + .5 * z.square().sum()

        d_j_z = torch.autograd.grad(j_log_lkhd, z)[0]
        z = z - 0.5 * (self.delta_1 ** 2) * d_j_z \
                + self.delta_1 * torch.randn_like(z)
        z = z.detach()
    return z
```

**Updating Model Parameters** Given the sampled latent variables **z**, we optimize the model parameters by minimizing the reconstruction error.

Listing 3: Updating model parameters.

```python
def update_G(im_t, fg, bg_shuf, pi_f, pi_b, bg,
             zf_logits, zb_logits):
    ### optimizers for generator networks
    fg_net_optimizer.zero_grad()
    bg_net_optimizer.zero_grad()
    sp_net_optimizer.zero_grad()
    ### optimizers for auxiliary classifiers
    fc_net_optimizer.zero_grad()
    bc_net_optimizer.zero_grad()

    ### Regularizations
    # Pseudo-label for additional regularization
    f_logits = fc_net(fg)
    b_logits = bc_net(bg)
    hpq_f = cross_ent(zf_logits, f_logits)
    hpq_b = cross_ent(zb_logits, b_logits)
    # orthogonal regularizations
    ortho_reg = orthogonal_reg(fg_net) + \
                    orthogonal_reg(bg_net)
    # TV-norm
    tv_norm = tv_loss(bg_shuf)

    ### log-lkhd for generators
    log_pf = - F.l1_loss(fg, im_t, reduction='none') \
             / (2. * SIGMA ** 2)
    log_pb = - F.l1_loss(bg_shuf, im_t, reduction='none') \
             / (2. * SIGMA ** 2)

    # posterior responsibility
    with torch.no_grad():
        ga_f = pi_f * log_pf.exp() \
            / (pi_f * log_pf.exp() + pi_b * log_pb.exp() + 1e-8)
    # objective for image generation
    e_z_log_p = ga_f.detach() * ((pi_f + 1e-8).log() + log_pf) \
        + (1. - ga_f.detach()) * ((pi_b + 1e-8).log() + log_pb)

    G_loss = - e_z_log_p.mean() + tv_norm * .01 + \
             hpq_f * .1 + hpq_b * .1 + ortho_reg * 1

    G_loss.backward()
    fg_net_optimizer.step()
    bg_net_optimizer.step()
    sp_net_optimizer.step()

    fc_net_optimizer.step()
    bc_net_optimizer.step()

def update_E(en_pos, en_neg):
    ebm_optimizer.zero_grad()

    ebm_loss = en_pos.mean() - en_neg.mean()
    ebm_loss.backward()

    ebm_optimizer.step()
```

```python
def learn(zp, zn, im_t):
    ### 1. Sampling latent variables
    zp = sample_langevin_posterior(zp)
    zn = sample_langevin_prior(zn)

    ### 2. Updating the parameters
    en_pos, zpf_logits, zpb_logits = ebm_net(zp)
    en_neg, znf_logits, znb_logits = ebm_net(zn)
    # update LEBMs
    update_E(en_pos, en_neg)

    im_p, fg, bg_shuf, pi_f, pi_b, bg = forward(zp)
    # update the generators
    update_G(im_t, fg, bg_shuf, pi_f, pi_b, bg,
             zf_logits, zb_logits)
```

# E   Evaluation Protocols

**Intersecion of Union (IoU)**   The IoU score measures the overlap of two regions $A$ and $B$ by calculating the ratio of intersection over union, according to

$$\text{IoU}(A, B) = \frac{|A \cap B|}{|A \cup B|}, \tag{12}$$

where we use the inferred mask and ground-truth mask as $A$ and $B$ respectively for evaluation.

**Dice (F1) score**   Similarly, the Dice (F1) score is

$$\text{Dice}(A, B) = \frac{2|A \cap B|}{|A| + |B|}. \tag{13}$$

Higher is better for both scores.

**Evalution**   As mentioned in the paper, IODINE [6] and Slot-attention [7] are designed for segmenting complex multi-object scenes using slot-based object representations. Ideally, the output of these models consists of masks for each individual object, while the background is viewed as a virtual "object" as well. In practice, however, it is possible that the model distributes the background over all the slots as mentioned in Locatello et al. [7]. Taking both cases into consideration (see Fig. 3 and Fig. 4), we propose two approaches to convert the multiple output masks into a foreground-background partition, and report the best results of these two options: (1) we compute the scores by making each mask as the background mask at a time, and then choose the best one; this works better when the background is treated as a virtual "object"; (2) we threshold and combine all the masks into a foreground mask; this is for when background is distributed to all slots.

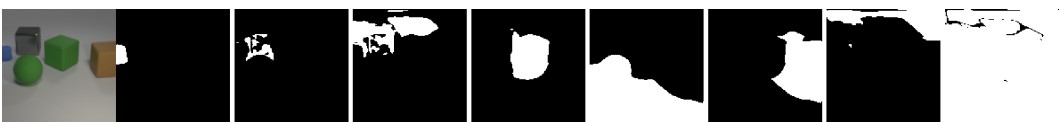

Figure 3: An example situation when using each individual mask as the background mask gives higher scores. Note that if we threshold the output of each individual slot and compose them, the result would be the mask shown in the last column.

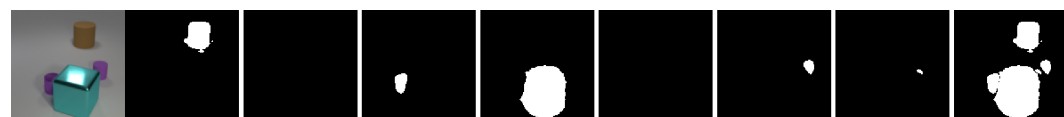

Figure 4: An example situation when thresholding&combining the output of each individual slot gives higher scores. We can see from the last column that the combined mask fits the foreground objects well.

# F    Additional Illustrations and Baseline Results

## F.1    More Examples

We provide more foreground extraction results of our model for each dataset; see Fig. 5, Fig. 6, Fig. 7 and Fig. 8. From top to bottom, we display: (i) observed images, (ii) generated images, (iii) masked generated foregrounds, (iv) generated backgrounds, (v) ground-truth foreground masks, and (vi) inferred foreground masks in each figure.

## F.2    Failure Modes

We provide examples for illustrating typical failure modes of the proposed model; see Fig. 9. On Birds dataset, we observe that the method can perform worse on samples where the foreground object has colors and textures quite similar to the background regions. Although the method can still capture the rough shape of the foreground object, some details can be missing. On TM-dSprites dataset, we observe that the method may occasionally miss one of the foreground objects. We conjecture that the problem can be mitigated with more powerful generator and further fine-tuning on this dataset.

## F.3    Baseline Results

**GrabCut**    We provide results of GrabCut [16] on Birds dataset and TM-dSprites dataset, shown in Fig. 10. We can see that GrabCut algorithm may fail when the foreground object and background region have moderately similar colors and textures. On TM-dSprites dataset, GrabCut algorithm outperforms other baselines, but is still inferior to the proposed method and exhibits a similar failure pattern.

**ReDO**    We provide results of ReDO [17] on Birds dataset and TM-dSprites dataset, shown in Fig. 11. ReDO overall performs better than GrabCut on Birds dataset, while it may fail when the background regions become more complex. We can also observe that ReDO relies heavily on the pixel intensities for foreground-background grouping on TM-dSprites dataset.

**IODINE**    On Birds dataset, we observe that IODINE [6] tends to use color as a strong cue for segmentation, see Fig. 12. On TM-dSprites dataset, IODINE is distracted by the background; see Fig. 13. These two findings are consistent with those reported in [6];

**Slot-Attention**    On Birds dataset, Slot-attention learns to roughly locate the position of foreground objects, but mostly fails to provide foreground masks when the background region becomes complex; see Fig. 14. Similarly, we can observe that Slot-Attention tends to use color as a strong cue for segmentation. On TM-dSprites dataset, Slot-attention is distracted by the background; see Fig. 15.

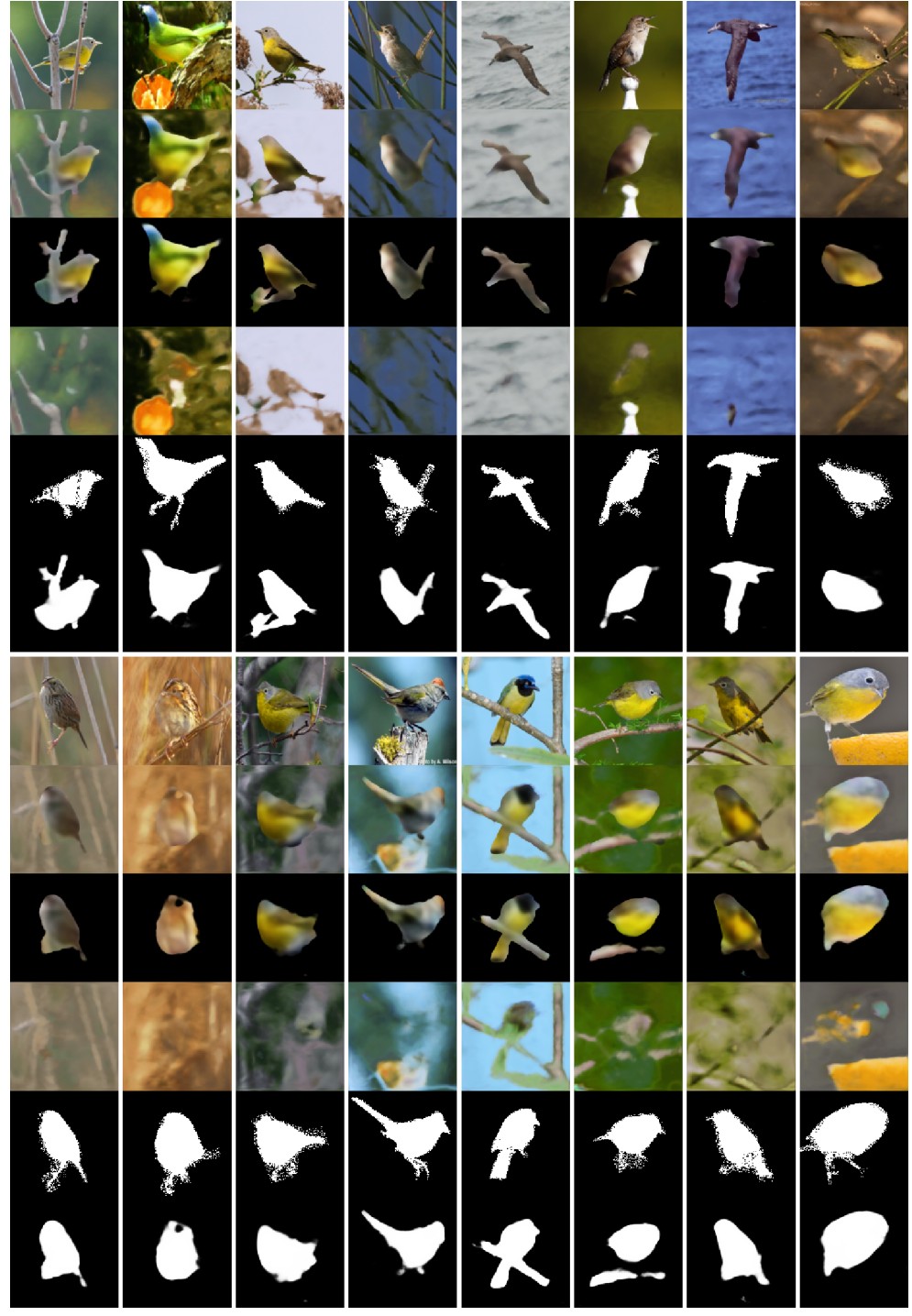

Figure 5: Additional foreground extraction results on Birds dataset.

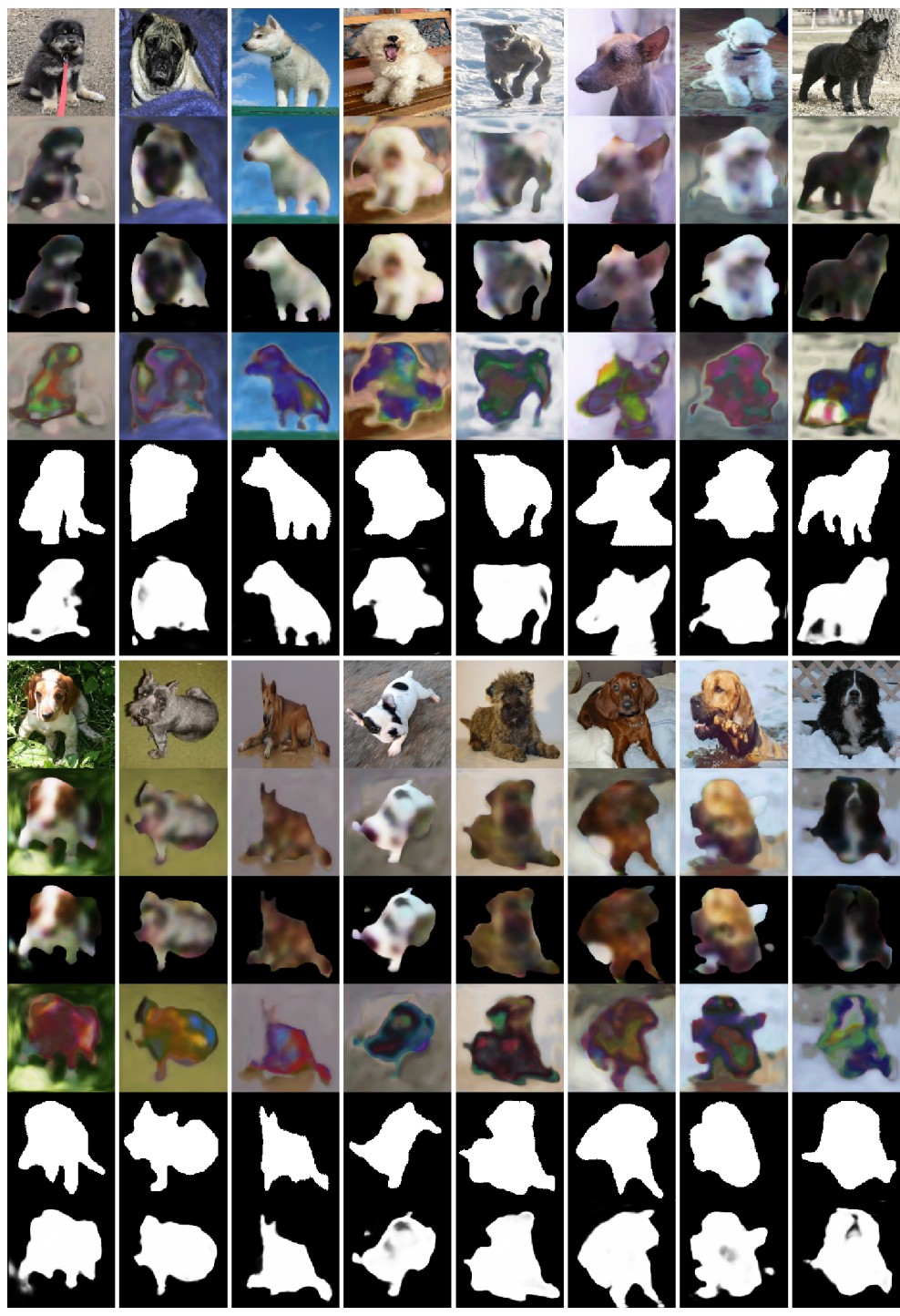

Figure 6: Additional foreground extraction results on Dogs dataset.

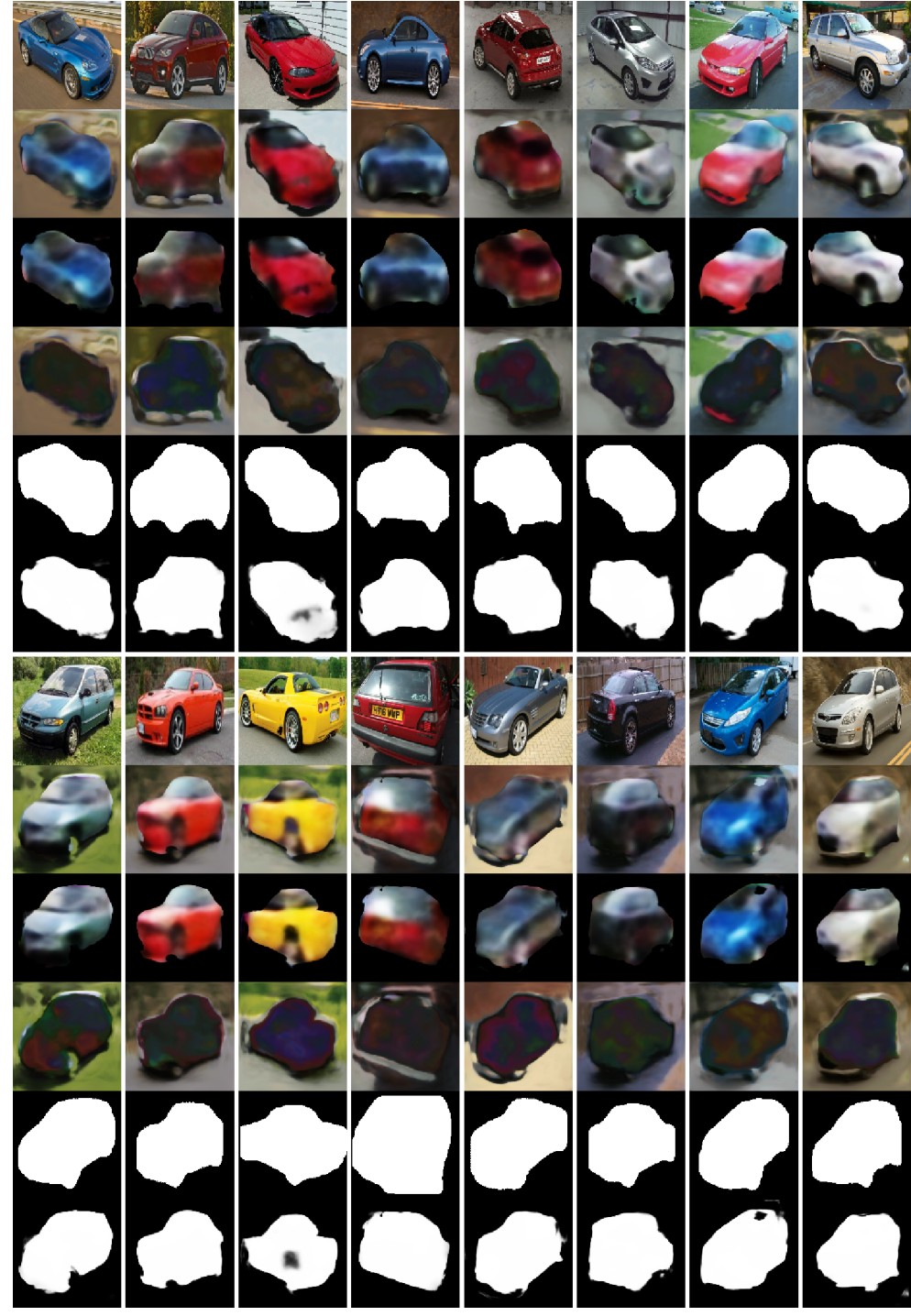

Figure 7: Additional foreground extraction results on Cars dataset.

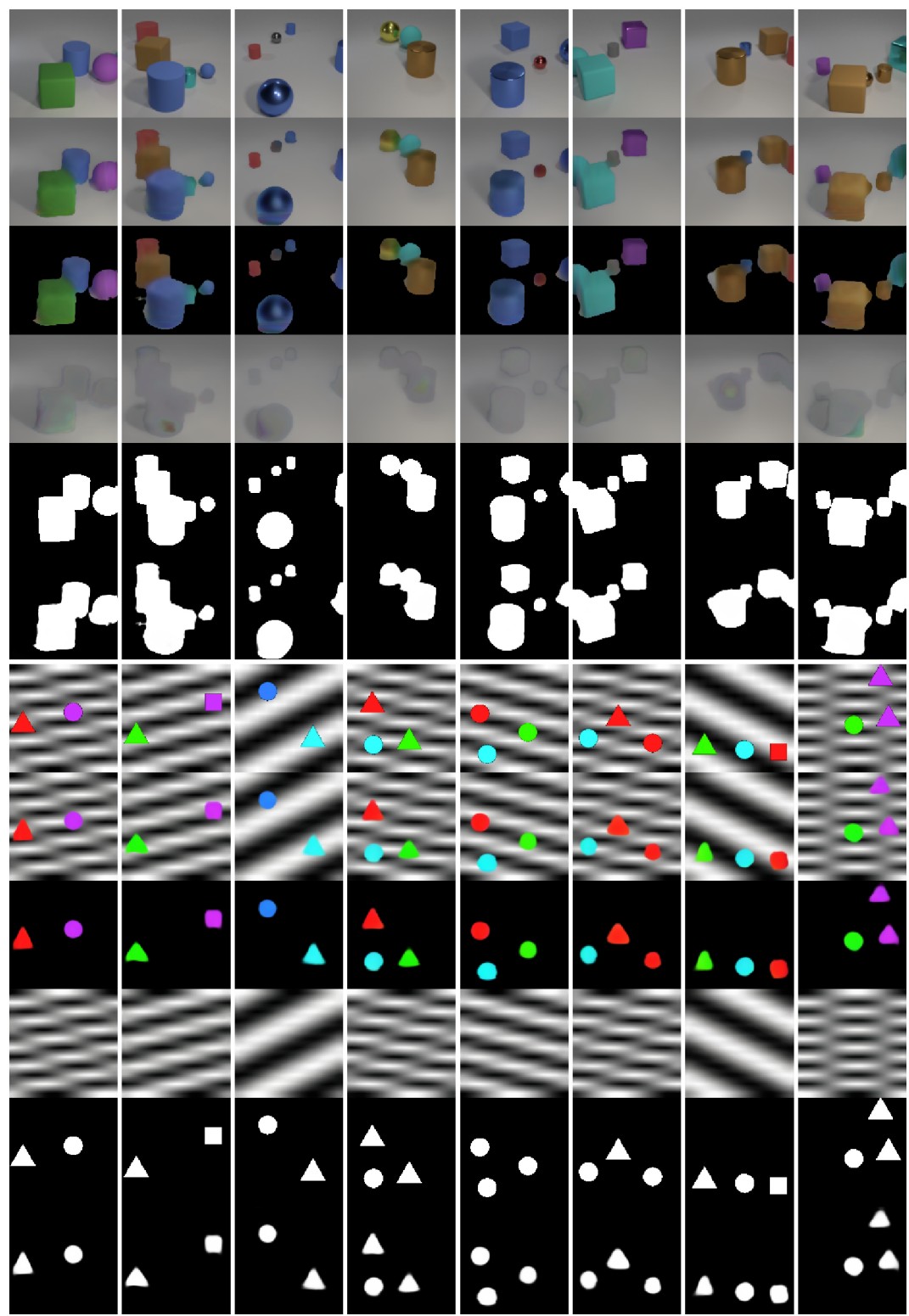

Figure 8: Additional foreground extraction results on CLEVR6 and TM-dSprites datasets.

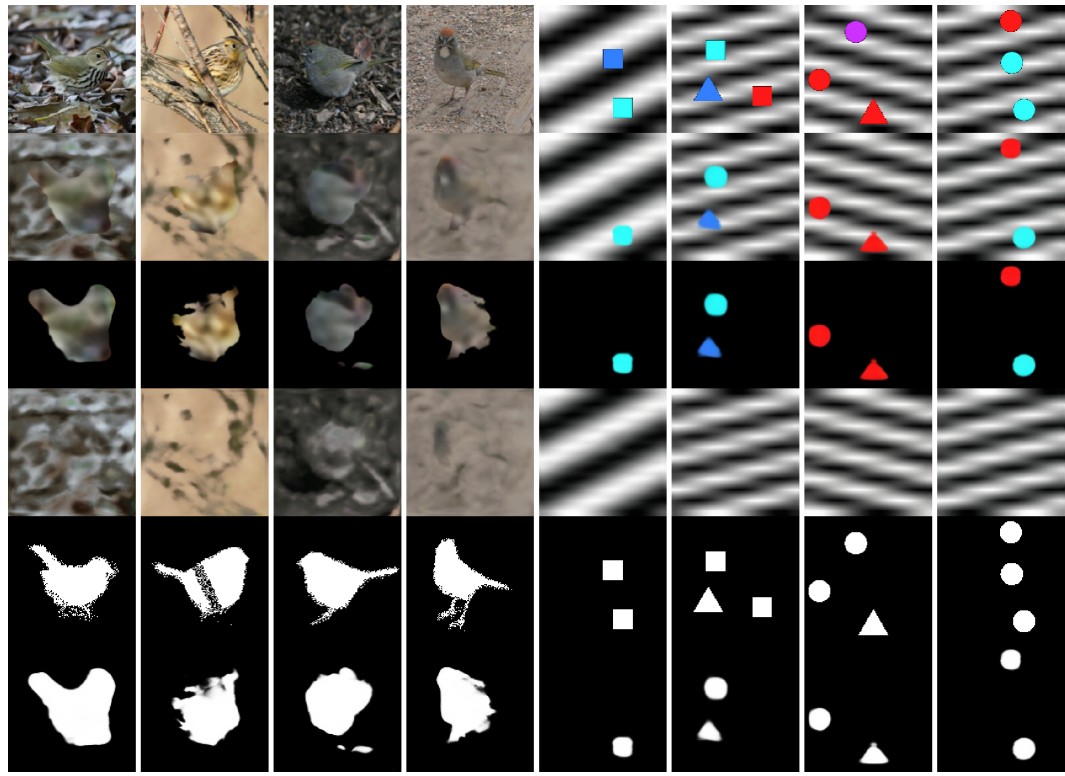

Figure 9: Typical failure modes on Birds and TM-dSprites.

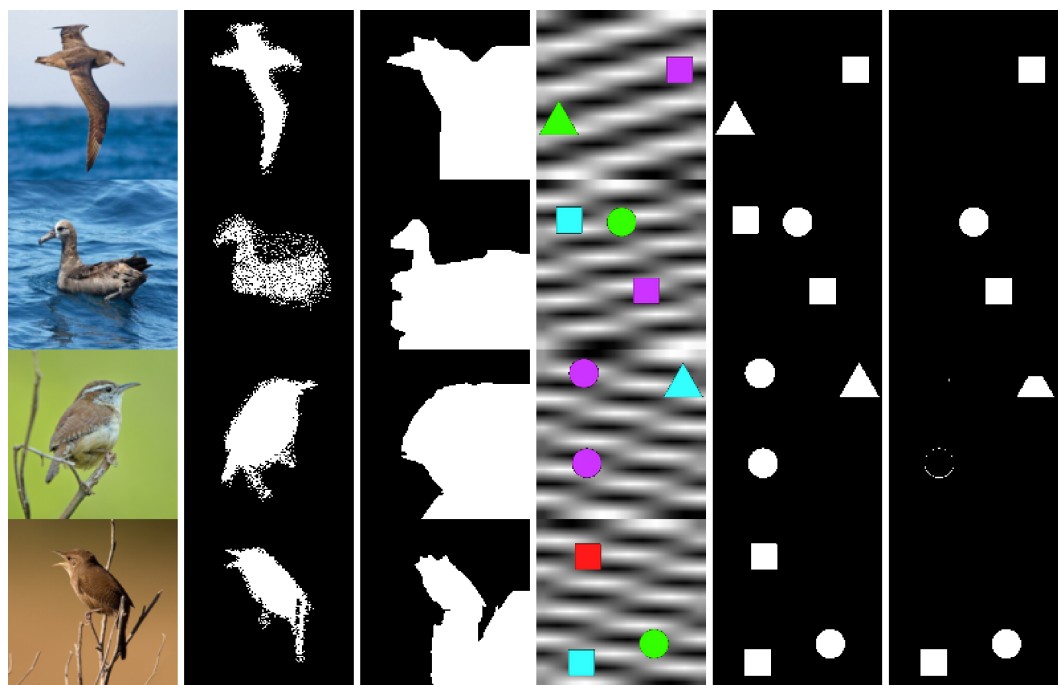

Figure 10: Results of GrabCut on Birds and TM-dSprites datasets. The first three columns are results from Birds dataset, and the last three are from TM-dSprites. From left to right, we display the observed image, ground-truth mask, and the foreground extraction results respectively.

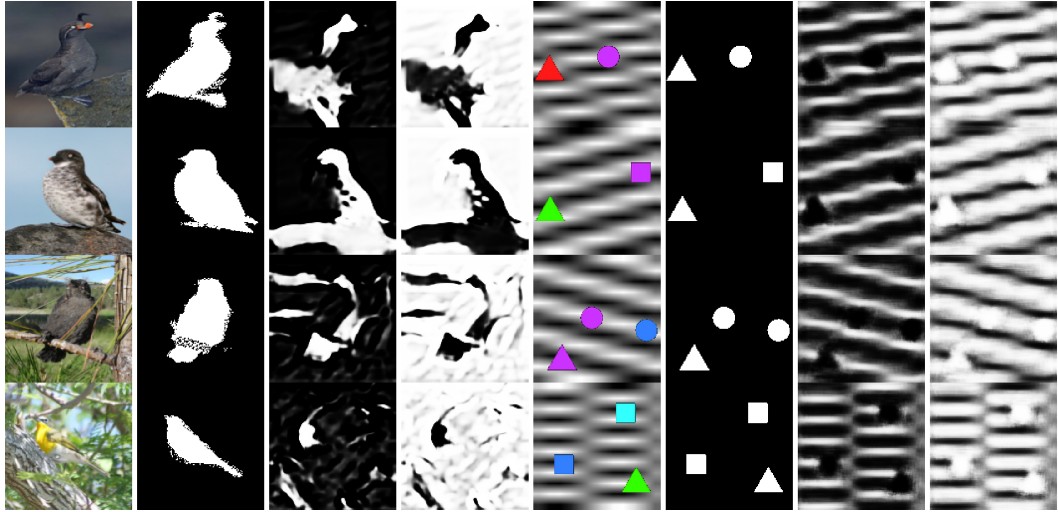

Figure 11: Results of ReDO on Birds and TM-dSprites datasets. The first four columns are results from Birds dataset, and the last four are from TM-dSprites. From left to right, we display the observed image, ground-truth mask, mask from the first output channel and from the second channel respectively.

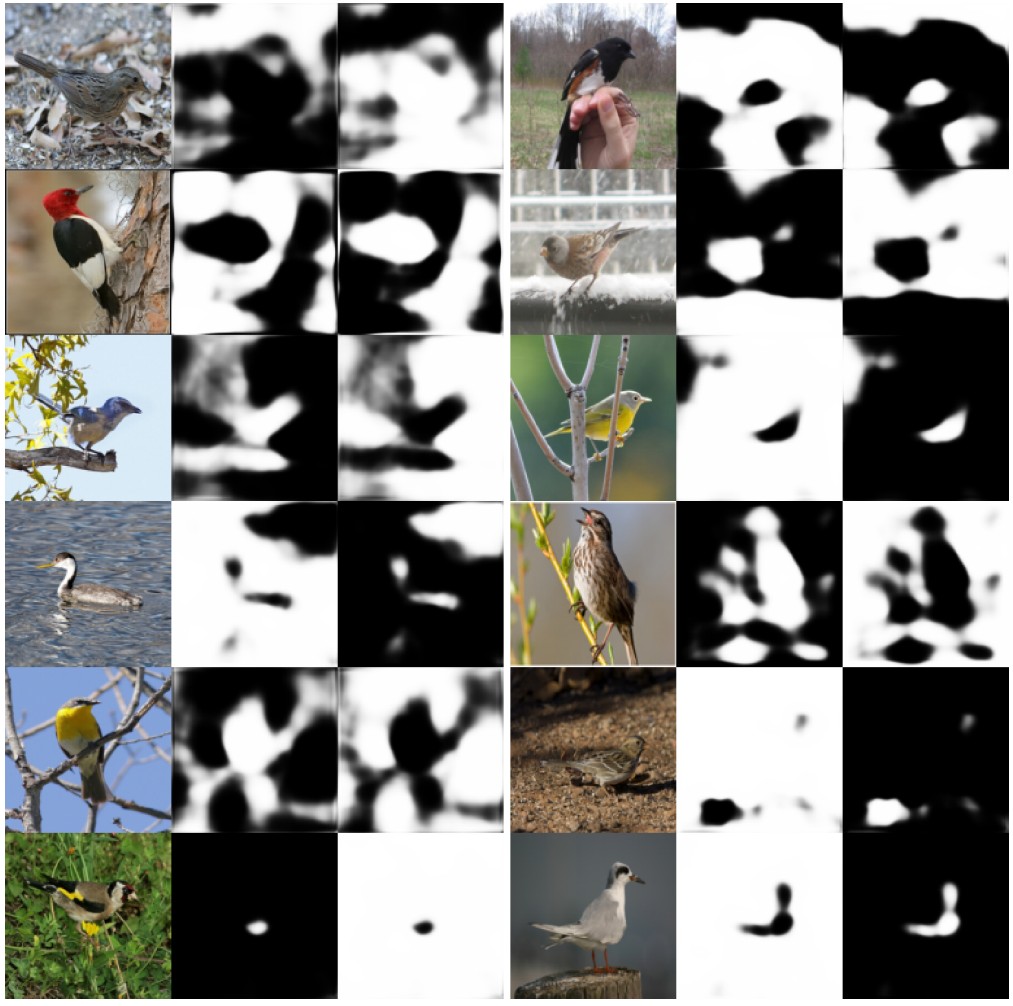

Figure 12: Results of IODINE on Birds datasets. We provide the observed image, mask from the first slot and from the second slot respectively.

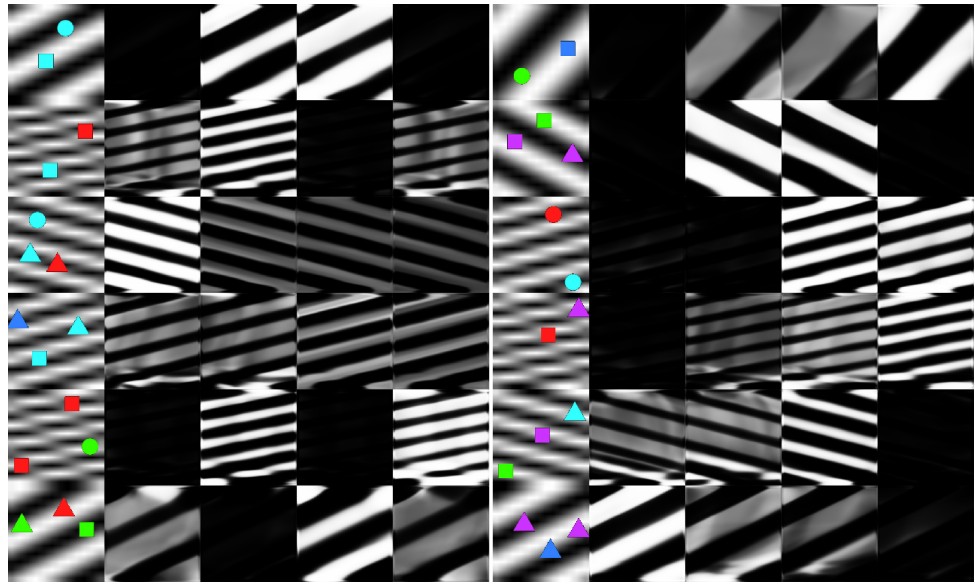

Figure 13: Results of IODINE on TM-dSprites datasets. We provide the observed image and masks from four object slots respectively.

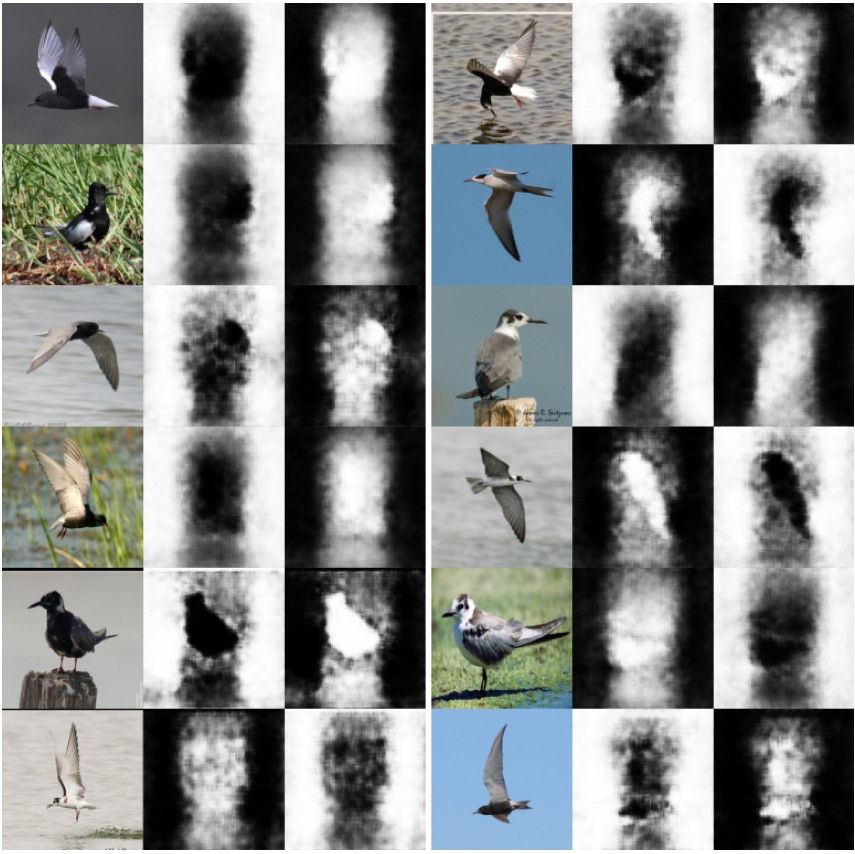

Figure 14: Results of Slot-Attention on Birds datasets. We provide the observed image, mask from the first slot and from the second slot respectively.

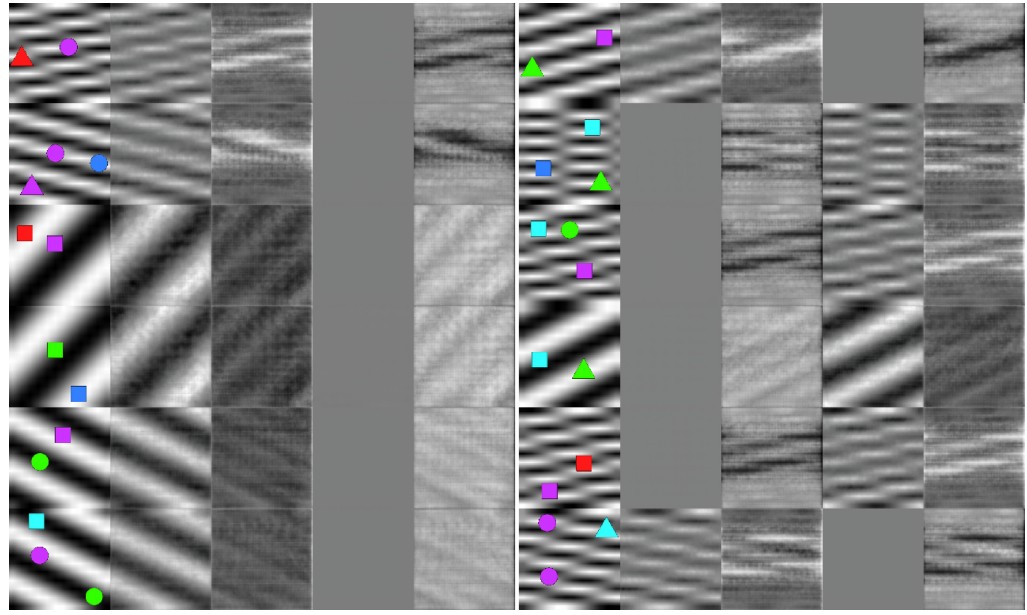

Figure 15: Results of Slot-Attention on TM-dSprites datasets. We provide the observed image and masks from four object slots respectively.

# G Further Discussion

## G.1 Preliminary Analysis of Real-world Datasets

We provide preliminary analysis of the statistics of the three real-world datasets. To measure the similarity of colors and textures for these datasets, we calculate the image histogram for the foreground objects and background regions of each dataset; see Fig. 16. To probe the similarity of shape distributions, we also provide the heatmap of foreground masks, as shown in Fig. 17. The heatmaps are calculated by overlapping the ground-truth masks and normalizing the summarized intensities with the maximum values. Despite the apparent difference in Birds vs Dogs and Cars, we can see that the data distribution of Birds dataset is more similar to that of Dogs dataset than to that of Cars dataset. We can also observe the similarity between the distributions of Dogs and Cars datasets. This could partly explain why the proposed method shows relatively strong performance on objects from unseen categories, *i.e.*, it effectively combines the colors, textures and shapes for foreground extraction.

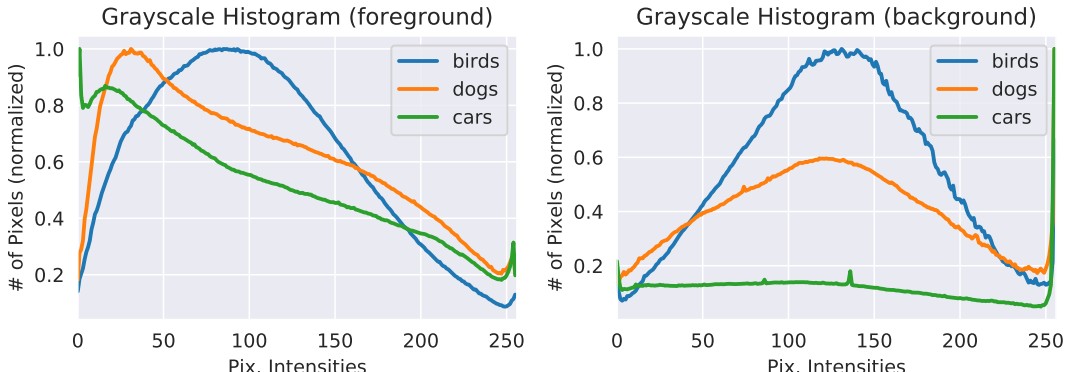

Figure 16: Image histograms for foreground objects and background regions from each dataset.

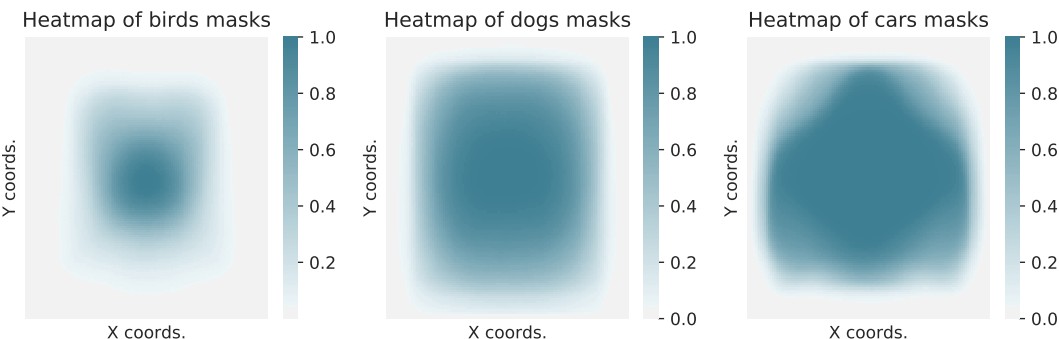

Figure 17: Heatmaps of ground-truth masks for each dataset.

## G.2 Possible Extension to Multi-Object Segmentation

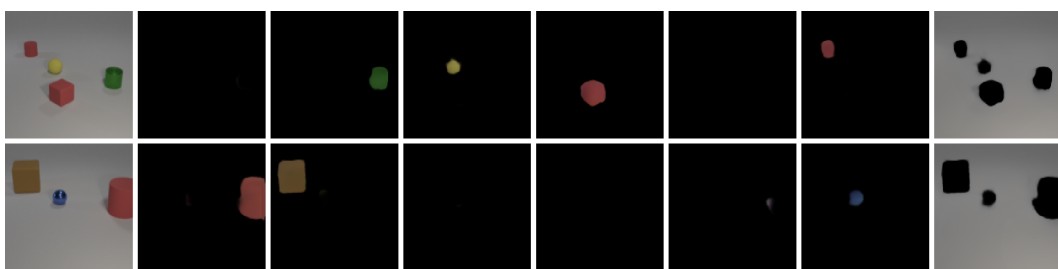

Figure 18: Preliminary results on learning slot-based object representation.

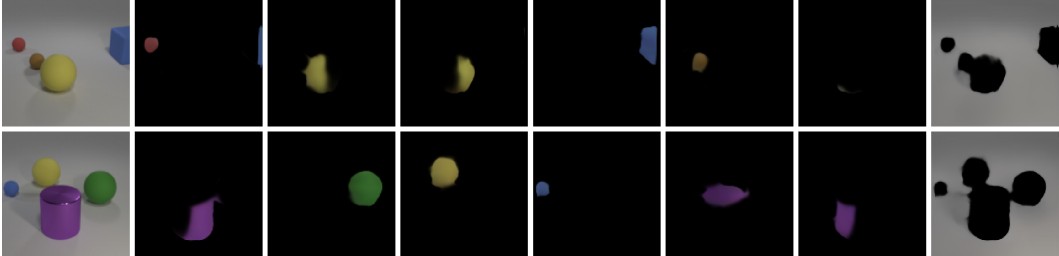

Figure 19: Failure modes of energy-based slot representation model.

We explore the possibility of using our model for segmenting and disentangling multiple objects. As shown in Fig. 18, the proposed method can disentangle the foreground objects, while providing explicit identification of the background region. However, we find that the model occasionally distributes a single object into several slots based on the difference in texture and shading; see Fig. 19. We conjecture that this is due to the lack of objectness modeling. We would like to investigate more on this direction in future works.

### G.3 Prior Sampling Results on Birds Dataset

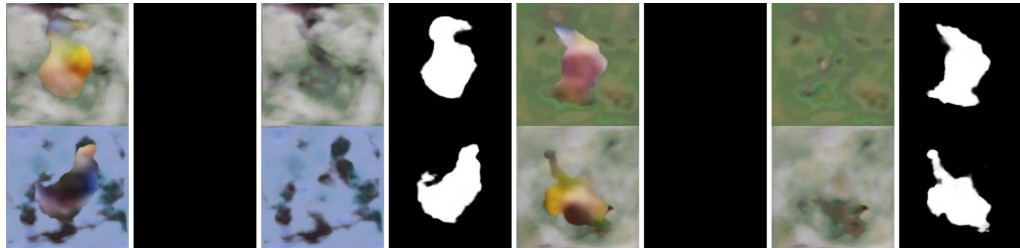

Figure 20: Prior sampling results on Birds dataset.

We provide preliminary results of sampling from the learned energy-based priors, as shown in Fig. 20. Of note, the generated prior samples are generally less realistic compared with the posterior samples, as prior sampling does not involve the region competition between foreground and background components, which may lead to worse separation and the generation of foreground and background regions. We would further explore generating foreground and background in future work.