# OpenReview forum: "Unsupervised Foreground Extraction via Deep Region Competition"
_NeurIPS.cc/2021/Conference — NeurIPS 2021 Poster_

### Official Review · Reviewer_rkT2 · 2021-07-16

**Rating:** 6
**Confidence:** 3

**Summary:**

This paper proposes deep region competition to extract foreground objects from images in an unsupervised manner. The proposed method leverages energy-based prior with generative image modeling in the form of Mixture of Experts. In addition, a pixel re-assignment scheme is further learned to capture the regularities of background regions. Experiments demonstrate that the proposed method can achieve promising results on real-world images.

**Limitations And Societal Impact:**

It seems that the proposed method can only work under single objects or multi-objects with clear backgrounds. It is not clear the performance of the proposed method on images in the wild. I suggest the authors to compare their methods with unsupervised salient object detection approaches to further verify their merits.

**Main Review:**

I think the proposed method is of some novelty and may be insightful for the community. It has also shown superior performance on 5 challenging datasets. My major concern on this paper is clarity. It is hard for me to fully understand the description. Instead of directly going into the technical details, I suggest the authors first provide a high-level overview of the task as well as the proposed method, and explain what are the important components of this method, what are their functions, how are they incorporated and learned? By doing so, the authors can further claim which part of the method is their unique design, and what are the advantages or differences to prior methods.

In the methodology sections, I do not understand how to achieve pixel re-assignment, and what is its relationship to the proposed method. In the experiment section, it is not clear what is the input resolution and how long does it take to train/test the proposed method.

The authors claim to provide limitations in section 4.1. I cannot find detailed discussions. Please provide in-depth discussions on the limitation of this work and potential solutions.

**Time Spent Reviewing:**

3

---

> ### Author Response · Authors · 2021-08-10
> **Thank you for your insightful comments**
>
> We sincerely thank you for your time and constructive comments. We hope our detailed responses below would resolve the remaining questions you have.
>
> **Further clarification of the task as well as the proposed method**
>
> **Task**
>
> As mentioned in the introduction section (L18-20), the goal of this paper is foreground extraction, a special case of generic image segmentation. Foreground extraction aims for a binary partition of the given image with specific semantic meaning, i.e., a foreground part that typically contains identifiable objects and the possibly less structural remaining regions as the background.
>
> **Method**
>
> - We propose to solve the foreground extraction problem by reconciling energy-based prior (LEBM) with generative image modeling in the form of Mixture of Experts (MoE) in this work. Similar to a VAE-based method, a latent-space vector needs to be sampled from the distribution and then transformed into an image by the decoder network. However, different from VAEs, the LEBM provides a learnable and more expressive prior distribution. Sampling from the distributions is realized by differentiable MCMC sampling techniques (Langevin dynamics [1] in this paper) instead of amortized inference, which leads to better posterior samples [2].
> - We further introduce the learned pixel re-assignment as the essential inductive bias to capture the background regularities. This can be viewed as an additional, spatial resampling step incorporated in the background generative model that consolidates the background statistics. Being able to identify the background component, intuitively, the model can potentially learn to extract the foreground part from the image.
> - We derive a principled learning objective and corresponding algorithm. The devised algorithm effectively exploits the interaction between the mixture components during the partitioning process, which resembles the intuition described in [3] (see section 3.2 L175-183 and supplemental material C.1 L81-99).
>
> **Understanding pixel re-assignment**
>
> >How to achieve pixel re-assignment?
>
> As described in section 3.1 L148-154, we use a separate pair of energy-based prior model and generative model to learn the re-assignment. We provide pytorch-style code in the supplemental material D L134-154 to facilitate understanding. In our implementation, the re-assignment function follows the output of a decoder network, which is a shuffling grid with the same size as the image. Its values indicate the "re-assigned" pixel coordinates. We aggregate the original background image pixels using these re-assigned pixel coordinates to perform the pixel re-assignment. The re-assignment model is jointly learned with the foreground and background model by optimizing the derived learning objective.
>
> >What is its relationship to the proposed method?
>
> The pixel re-assignment step can be viewed as a spatial re-shuffling of the original pixels generated by the background decoder. This key inductive bias is introduced to capture the background regularities in the images and help the model to identify the background component.
>
> **Clarification of the experiment details**
> >It is not clear what the input resolution is.
>
> The input image size is $128\times 128$. We provide more details about the data curation in supplemental material A.
>
> >How long does it take to train/test the proposed method.
>
> As mentioned in supplemental material B L71-73, we run experiments on a single V100 GPU with 16GB RAM and with a batch size of 48. We set the maximum training iterations to 10K and run for at most 48 hours for each dataset. Although our method involving MCMC sampling is more costly than the encoder networks in Slot Attention models, it is comparable to the amortized iterative inference in IODINE.
>
> **In-depth discussions on the limitation of this work and potential solutions**
>
> We summarize the limitations of our work as follows: (a) As mentioned in section 4.1, the fidelity of the reconstructed foreground and background regions can be further improved. DRC reconstructs foreground and background regions less realistic than those reconstructed by state-of-the-art GANs. (b) As shown in the supplemental material G.2, the modeling approach in its current form is not able to provide instance-level segmentation of multiple objects well. (c) We provide typical failure cases in supplemental material F.2. We observe that the performance degrades on samples where the foreground object has colors and textures quite similar to the background regions.
>
> One possible solution to improve the reconstruction quality is to learn a LEBM by diffusion recovery likelihood [4]. By considering the noise level during inference and maximizing the corresponding conditional probability, the LEBM may achieve better sampling quality, and consequently better generation and reconstruction quality. In addition, evidence in [2] has shown that larger LEBMs result in better reconstruction quality as well.
>
> As for the instance-level segmentation of multiple objects, we observe that the proposed method can disentangle the foreground objects, providing explicit identification of the background region, while occasionally distribute a single object into several slots based on the difference in texture and shading. Following your suggestion, we have a closer look in the generated mask and reconstructed image in each slot, and find an interesting phenomenon -- It is normal for both the proposed model and the baselines (IODINE and slot attention) to have several slots assigned to the same object when the number of objects in the scene is smaller than the fixed number of slots in the model (see Figure 19 in the supplemental material). Since all these generative models are Gaussian mixtures, which mix reconstructed images with masks, the model can (1) either assign different parts of an object into some highly confident slots (masks with prob. close to 1), or (2) assign the same information to all slots with 1/N weighted mask each. The proposed model differs from the baseline in that all slots are quite certain about their responsibility, and thus have 0.9 weighted masks; while masks in baseline models are more uncertain. Intuitively, both of these are reasonable. We conjecture that the long-run MCMC sampling in our model induces over-saturation in masks, as discovered in [5]. It would be interesting to follow the solution mentioned in [5] in future work, i.e., using correct tuning of Langevin noise.
>
> **Comparison with unsupervised salient object detection approaches**
>
> We have added three unsupervised salient object detection approaches as our baselines. The results are shown in the table below. We observe that our method outperforms the unsupervised salient object detection baselines, especially on the TM-dSprites dataset with strongly confounding background parts, which demonstrates the efficacy of the proposed method.
>
> | | Birds | Dogs | Cars | CLEVR6 | TM-dSprites
> |:--:   | :--:   | :--:   | :--:  | :--:    | :--: |
> | |IoU/Dice|IoU/Dice|IoU/Dice|IoU/Dice|IoU/Dice|
> |[6]   | 52.3/66.8| 56.1/69.7| 48.2/63.0| 53.5/66.9| 18.6/27.9 |
> |[7]   | 24.7/37.9| 34.5/50.5| 33.4/49.5| 41.9/56.7|  1.3/2.5 |
> |[8]   | 24.0/37.0| 20.0/32.5| 24.7/39.2| 19.3/31.4| 21.4/34.5 |
> |Ours  |**56.4**/**70.9** | **71.7**/**83.2** | **72.4**/**83.7** | **84.7**/**91.5** | **78.8**/**87.5**|
>
> [1] Max Welling and Yee W Teh. Bayesian learning via stochastic gradient langevin dynamics. In Proceedings of International Conference on Machine Learning (ICML), 2011.
>
> [2] Bo Pang, Tian Han, Erik Nijkamp, Song-Chun Zhu, and Ying Nian Wu. Learning latent space energy-based prior model. In Proceedings of Advances in Neural Information Processing Systems (NeurIPS), 2020.
>
> [3] Song Chun Zhu and Alan Yuille. Region competition: Unifying snakes, region growing, and bayes/mdl for multiband image segmentation. IEEE Transactions on Pattern Analysis and Machine Intelligence (TPAMI), 18(9):884–900, 1996.
>
> [4] Ruiqi Gao, Yang Song, Ben Poole, Ying Nian Wu, and Diederik P Kingma. Learning energy-based models by diffusion recovery likelihood. In International Conference on Learning Representations (ICLR), 2020.
>
> [5] Erik Nijkamp, Mitch Hill, Tian Han, Song-Chun Zhu, and Ying Nian Wu. On the anatomy of mcmc-based maximum likelihood learning of energy-based models. In Proceedings of AAAI Conference on Artificial Intelligence (AAAI), 2020.
>
> [6] Ming-Ming Cheng, Niloy J Mitra, Xiaolei Huang, Philip HS Torr, and Shi-Min Hu. Global contrast based salient region detection. IEEE Transactions on Pattern Analysis and Machine Intelligence (TPAMI), 37(3):569–582, 2014.
>
> [7] Sebastian Montabone and Alvaro Soto. Human detection using a mobile platform and novel features derived from a visual saliency mechanism. Image and Vision Computing, 28(3):391–402, 2010.
>
> [8] Xiaodi Hou and Liqing Zhang. Saliency detection: A spectral residual approach. In Proceedings of the IEEE Conference on Computer Vision and Pattern Recognition (CVPR), 2007.

---

### Official Review · Reviewer_RcgA · 2021-07-16

**Rating:** 7
**Confidence:** 4

**Summary:**

In this work, the authors propose an unsupervised, generative model for foreground-background segmentation of RGB images which is based on the previously proposed "Latent Space Engergy-Based Prior Model" (LEBM). Similar to a VAE, a latent vector sampled from a prior distribution is transformed into an image by a decoder network. Differently from VAEs however, the LEBM uses a more complex, learnable prior distribution and sampling from the prior and posterior distributions is realized by differentiable MCMC sampling using Langevin dynamics. An image is modelled using a spatial mixture of foreground and background images decoded by separate decoder networks. To foster the segregation into foreground and background, the backround model includes an additional, spatial resampling step targeted at the assumed background statistics. While the reconstructed images tend to be blurry, the reconstructed foreground-background masks have been shown to be more accurate than previous methods on several datasets.

**Limitations And Societal Impact:**

- The authors do not discuss the limitations of the proposed model, e.g. the necessary preprocessing of the images or conditions in which the background prior might fail.
- There is no discussion of the potential societal impact included in the paper.

**Main Review:**

### Originality
- This work is an application of an existing modeling technique (LEBM) to an existing task (unsupervised foreground segmentation). To my knowledge, the combination however is novel.
- It is made clear in the paper, to which degree the approach is adapted from previous work and which components are contributed with this work.
- The paper does not contain a related work section. Instead, some relations to previous work are discussed in the introduction, and some in the method section (L184-191). In my view, discussing previous works to provide context for the current work is an integral part of every scientific article and deserves to be discussed in a separate section with greater detail. I am happy to improve my rating of the paper if this is addressed. (**Update:** This has been well addressed by the authors.)

### Quality
- The derivation of the model and its training procedure is described in great detail, and technically sound to me. The information given in the main paper is sufficient to comprehend the model and experiments, further details about the datasets, hyperparameters and model training are provided in-depth in the supplemental information.
- The model is compared with previous approaches on five datasets, including datasets with natural images. An ablation study is performed to judge the contribution of every modeling component. In the main results on the segmentation performance (Table 1), the models are however evaluated on the training set. Only in Table 3 when evaluating the generalizability, a subset of the models is evaluated on a separate test set for a subset of the datasets. In my view, testing on held out samples should also be the norm for unsupervised methods, since overfitting might also happen in this case. Ideally, the same information would be given in Table 1, but using separate test sets. If evaluating on the training set is required for comparing to previous methods, this should be made explicit and featured less prominently.
  (**Update**: This has been addressed by the authors
- The paper only shows and evaluates reconstructions (including the segmentation mask) of images, but no samples from the model. While not strictly necessary for the target task of foreground-background segmentation, in my view this is still necessary to fully understand the modelling approach in this setting. The authors propose a generative model, which is derived with the goal of fitting the data distribution, but do not directly evaluate to which degree the model succeeds in that respect. Evaluating and/or showing samples from the model would be a means to obtain more insights into this.

### Clarity
- The paper and supplement are very well written. The derivation of the model and the experiments are well described, and the necessary concepts from previous works briefly introduced. My main criticism is about the missing related work section (see above).

### Significance
- Foreground-background segmentation is a classical task in computer vision with several applications, and using unsupervised models has many practical advantages. This model is therefore relevant to both researchers and practitioners in that field. Compared to more general segmentation approaches as, e.g. semantic or instance segmentation, the applicability of the model is limited in practice. From the perspective of applying this method, the paper would profit from a discussion of the real world relevance of unsupervised foreground-background segmentation in the introduction.
- Interest in unsupervised methods in computer vision has increased in recent years. Structured latent-space models are one particular area with much recent interest, often under the umbrella term "object-centric representation learning". Mostly this is approached by using GAN or VAE-based models. The sucessful demonstration of a different generative method for foreground-background modelling is very interesting and might well inspire further research in that field. From the perspective of this field, the present work however has two important limitations: (1) As shown in the supplemental material, the modelling approach in its current form is not able to model multiple objects well, which is one of the core challenges in that field. (2) The model is not able to handle appearances well, as opposed to the masks.


### Update
Overall my concerns have been addressed, especially regarding the related work. Therefore I improve my rating of this paper accordingly. In summary I think this is a good paper, however I still see some minor points where this paper could be improved further.

**Testing on training data.** This decision is justified to be able to compare to previous methods, and the proposed model is additionally tested on held-out data. In my view the paper could however be improved further by restructuring the result presentation to push more towards proper evaluation.

**Testing generative modeling capabilities.** I still think it would have been interesting to go more into depth regarding the generative modeling capabilities of the model. Since the main focus of this work is foreground-background segmentation, leaving this for future work is however justified.

**Time Spent Reviewing:**

6

---

> ### Author Response · Authors · 2021-08-10
> **Thank you for your insightful comments**
>
> We sincerely thank you for your time and thoughtful comments. We provide point-to-point responses to address the questions that hopefully would increase your rating of our work.
>
> **Related works in a separate section with greater detail**
>
> We have added a more detailed related works section with the main body as follows:
>
> A typical line of methods frames unsupervised or weakly supervised foreground segmentation within a generative modeling context. Several methods build upon generative adversarial networks (GAN) [1] to perform foreground segmentation. LR-GAN [2] learns to generate background regions and foreground objects separately and recursively, which simultaneously produces the foreground objects mask. ReDO (ReDrawing of Objects) [3] proposes a GAN-based object segmentation model, based on the assumption that replacing the foreground object in the image with a generated one does not impact the distribution of the training data, given that the foreground object is correctly discovered. Similarly, SEIGAN [4] learns to extract foreground objects by recombining the foreground objects with the generated background regions. FineGAN [5] generates images in a hierarchical manner (i.e., first specifying the object shape and then the object texture) to disentangle the background and foreground object. Benny and Wolf [6] further hypothesize that solving an ensemble of unsupervised tasks together enables the model to improve on the performance compared with the one that solves each individually. They, therefore, train a complex GAN-based model (OneGAN) to solve several tasks simultaneously, including foreground segmentation. Although LR-GAN and FineGAN do produce masks as part of their generative process, they cannot segment a given image. SEIGAN and OneGAN achieve decent performance on foreground-background segmentation, while these methods require a set of clean background images as additional inputs for weak supervision. ReDO capture the foreground objectness with possibly oversimplified assumptions, which could potentially limit its application to datasets where object shapes vary significantly. On another front, compositional generative scene models [7–15], sharing the idea of scene decomposition stemming from DRAW [16], learn to represent foreground objects and background regions in terms of a collection of latent variables with the same representational format. These methods typically exploit the spatial mixture model for generative modeling. IODINE [8] proposes a slot-based object representation method and models the latent space using iterative amortized inference [17]. Slot-Attention [9], as a step forward, effectively incorporates the attention mechanism into slot-based object representation for flexible foreground object binding. Both methods use fully shared parameters among individual mixture components to entail permutation invariance of the learned multi-object representation. Alternative models such as MONet [7] and GENESIS [14] use multiple encode-decode steps to perform scene decomposition and foreground object extraction. Although these methods exhibit strong performance on synthetic multi-object datasets with simple background and foreground shapes, they may fail when dealing with complex real-world data or even synthetic datasets with more challenging background [8,9].
>
> More closely related to the classical methods, another line of work focuses on utilizing image features extracted by deep neural networks, or designing energy functions based on data-driven methods to define the desired property of foreground objects. [18] and [19] obtain impressive results when depth images are accessible in addition to usual RGB images, while such methods are not directly applicable for pure RGB images. W-Net [20] extracts image features via a deep auto-encoder jointly trained by minimizing reconstruction error and normalized cut. The learned features are further processed by CRF smoothing to perform hierarchical segmentation. Kanezaki [21] proposes to employ a neural network as part of the partitioning criterion (inspired by [22]) to directly minimize the chosen intra-region pixel distance for segmentation. Ji et al. [23] propose to use Invariant Information Clustering as the objective for segmentation, where the network is trained to be part of the learned distance. IEM [24] performs unsupervised segmentation by partitioning images into maximally independent sets, with the goal of minimizing the predictability of one set from the other. This line of works are generally bottle-necked by the selected post-processing segmentation algorithm, or require extra transformations to produce meaningful foreground segmentation masks. Apart from these lines of methods, additional efforts have also been devoted with weak supervision using image classification labels [25-27], bounding boxes [28,29], or saliency maps [30,31] to perform foreground-background segmentation.
>
> We are willing to include additional related literature in case we miss something. We welcome your suggestions.
>
> **Emphasizing the evaluation of generalizability**
>
> Thank you for your advice. We would point out explicitly in the caption of Table 1 that the evaluation is conducted on the training set following the previous methods. We agree that testing on held-out samples should also be the norm for unsupervised methods and have provided the results on unseen instances/categories in Tables 3 and 4. We would add the held-out testing results on CLEVR6 and TM-dSprites datasets in the supplemental material in revision.
>
> **Showing samples from the generative model**
>
> Thank you for your advice. We would add preliminary results of sampling from the learned energy-based priors in the supplemental material in revision. Of note, the generated prior samples are generally less realistic compared with the posterior samples, as prior sampling does not involve the region competition between foreground and background components, which may lead to worse separation and the generation of foreground and background regions. We would further explore generating foreground and background in future work.
>
> **Discussion of the real-world relevance in the introduction**
>
> Thank you for your advice. We would add a discussion for the real-world relevance of unsupervised foreground-background segmentation in the introduction. We think one of the possible applications for our method is to generate segmentation masks for unlabeled data. This may help to deal with problems in which labeled data is scarce or expensive to obtain.
>
> **The necessary pre-processing of the images**
>
> Thanks for pointing it out. Most pre-processing techniques in this work follow the standard practice. They are detailed in supplemental material A. The original images are center-cropped and then resized to $128\times 128$. In addition, we normalize the pixel value of the images from [0, 255] to [0, 1]. We subtract a pre-assumed mean value of 0.5 from the images and divide the normalized pixel values by 0.5 before feeding the images into the model.
>
> **Discussion of the conditions in which the background prior might fail**
>
> We will follow your advice to add an extended discussion of the limitations. We believe that the proposed modeling approach for the background part is generic. We have evaluated our method on real-world datasets with complex backgrounds and synthetic datasets with simple, nearly uniform-colored backgrounds and textured, and highly confounding backgrounds. We agree that our method still has its limitation. The performance may degrade when the foreground object has colors and textures quite similar to the background regions, as mentioned in the supplemental material F.2. We think it would be an interesting direction for future research.
>
> **Discussion of the potential societal impact**
>
> We think this paper proposes a general-purpose method. The proposed method can potentially help to deal with segmentation problems in which labeled data is scarce or expensive to obtain.

---

> > ### Author Response · Authors · 2021-08-10
> > **References**
> >
> > [1] Ian J Goodfellow, Jean Pouget-Abadie, Mehdi Mirza, Bing Xu, David Warde-Farley, Sherjil Ozair, Aaron C Courville, and Yoshua Bengio. Generative adversarial nets. In Proceedings of Advances in Neural Information Processing Systems (NeurIPS), 2014.
> >
> > [2] Jianwei Yang, Anitha Kannan, Dhruv Batra, and Devi Parikh. Lr-gan: Layered recursive generative adversarial networks for image generation. In International Conference on Learning Representations (ICLR), 2017.
> >
> > [3] Mickaël Chen, Thierry Artières, and Ludovic Denoyer. Unsupervised object segmentation by redrawing. In Proceedings of Advances in Neural Information Processing Systems (NeurIPS), 2019.
> >
> > [4] Pavel Ostyakov, Roman Suvorov, Elizaveta Logacheva, Oleg Khomenko, and Sergey I Nikolenko. Seigan: Towards compositional image generation by simultaneously learning to segment, enhance, and inpaint. arXiv preprint arXiv:1811.07630, 2018.
> >
> > [5] Krishna Kumar Singh, Utkarsh Ojha, and Yong Jae Lee. Finegan: Unsupervised hierarchical disentanglement for fine-grained object generation and discovery. In Proceedings of the IEEE Conference on Computer Vision and Pattern Recognition (CVPR), 2019.
> >
> > [6] Yaniv Benny and Lior Wolf. Onegan: Simultaneous unsupervised learning of conditional image generation, foreground segmentation, and fine-grained clustering. In Proceedings of European Conference on Computer Vision (ECCV), 2020.
> >
> > [7] Christopher P Burgess, Loic Matthey, Nicholas Watters, Rishabh Kabra, Irina Higgins, Matt Botvinick, and Alexander Lerchner. Monet: Unsupervised scene decomposition and representation. arXiv preprint arXiv:1901.11390, 2019.
> >
> > [8] Klaus Greff, Raphaël Lopez Kaufman, Rishabh Kabra, Nick Watters, Christopher Burgess, Daniel Zoran, Loic Matthey, Matthew Botvinick, and Alexander Lerchner. Multi-object representation learning with iterative variational inference. In Proceedings of International Conference on Machine Learning (ICML), 2019.
> >
> > [9] Francesco Locatello, Dirk Weissenborn, Thomas Unterthiner, Aravindh Mahendran, Georg Heigold, Jakob Uszkoreit, Alexey Dosovitskiy, and Thomas Kipf. Object-centric learning with slot attention. In Proceedings of Advances in Neural Information Processing Systems (NeurIPS), 2020.
> >
> > [10] Klaus Greff, Antti Rasmus, Mathias Berglund, Tele Hotloo Hao, Jürgen Schmidhuber, and Harri Valpola. Tagger: Deep unsupervised perceptual grouping. In Proceedings of Advances in Neural Information Processing Systems (NeurIPS), 2016.
> >
> > [11] Klaus Greff, Sjoerd van Steenkiste, and Jürgen Schmidhuber. Neural expectation maximization. In Proceedings of Advances in Neural Information Processing Systems (NeurIPS), 2017.
> >
> > [12] Sjoerd van Steenkiste, Michael Chang, Klaus Greff, and Jürgen Schmidhuber. Relational neural expectation maximization: Unsupervised discovery of objects and their interactions. In International Conference on Learning Representations (ICLR), 2018.
> >
> > [13] SM Ali Eslami, Nicolas Heess, Theophane Weber, Yuval Tassa, David Szepesvari, Koray Kavukcuoglu, and Geoffrey E Hinton. Attend, infer, repeat: Fast scene understanding with generative models. In Proceedings of Advances in Neural Information Processing Systems (NeurIPS), 2016.
> >
> > [14] Martin Engelcke, Adam R Kosiorek, Oiwi Parker Jones, and Ingmar Posner. Genesis: Generative scene inference and sampling with object-centric latent representations. In International Conference on Learning Representations (ICLR), 2020.
> >
> > [15] Zhixuan Lin, Yi-Fu Wu, Skand Vishwanath Peri, Weihao Sun, Gautam Singh, Fei Deng, Jindong Jiang, and Sungjin Ahn. Space: Unsupervised object-oriented scene representation via spatial attention and decomposition. In International Conference on Learning Representations (ICLR), 2020.
> >
> > [16] Karol Gregor, Ivo Danihelka, Alex Graves, Danilo Rezende, and Daan Wierstra. Draw: A recurrent neural network for image generation. In Proceedings of International Conference on Machine Learning (ICML), 2015.
> >
> > [17] Joe Marino, Yisong Yue, and Stephan Mandt. Iterative amortized inference. In Proceedings of International Conference on Machine Learning (ICML), 2018.
> >
> > [18] Trung T Pham, Thanh-Toan Do, Niko Sünderhauf, and Ian Reid. Scenecut: Joint geometric and object segmentation for indoor scenes. In Proceedings of International Conference on Robotics and Automation (ICRA), 2018.
> >
> > [19] Nathan Silberman, Derek Hoiem, Pushmeet Kohli, and Rob Fergus. Indoor segmentation and support inference from rgbd images. In Proceedings of European Conference on Computer Vision (ECCV), 2012.
> >
> > [20] Xide Xia and Brian Kulis. W-net: A deep model for fully unsupervised image segmentation. arXiv preprint arXiv:1711.08506, 2017.
> >
> > [21] Asako Kanezaki. Unsupervised image segmentation by backpropagation. In International Conference on Acoustics, Speech and Signal Processing (ICASSP), 2018.
> >
> > [22] D Ulyanov, A Vedaldi, and V Lempitsky. Deep image prior. International Journal of Computer Vision (IJCV), 128(7), 2020.
> >
> > [23] Xu Ji, João F Henriques, and Andrea Vedaldi. Invariant information clustering for unsupervised image classification and segmentation. In Proceedings of International Conference on Computer Vision (ICCV), 2019.
> >
> > [24] Pedro Savarese, Sunnie SY Kim, Michael Maire, Greg Shakhnarovich, and David McAllester. Information-theoretic segmentation by inpainting error maximization. In Proceedings of the IEEE Conference on Computer Vision and Pattern Recognition (CVPR), 2021.
> >
> > [25] George Papandreou, Liang-Chieh Chen, Kevin P Murphy, and Alan L Yuille. Weakly-and semi-supervised learning of a deep convolutional network for semantic image segmentation. In Proceedings of International Conference on Computer Vision (ICCV), 2015.
> >
> > [26] Deepak Pathak, Philipp Krahenbuhl, and Trevor Darrell. Constrained convolutional neural networks for weakly supervised segmentation. In Proceedings of International Conference on Computer Vision (ICCV), 2015.
> >
> > [27] Zilong Huang, Xinggang Wang, Jiasi Wang, Wenyu Liu, and Jingdong Wang. Weakly-supervised semantic segmentation network with deep seeded region growing. In Proceedings of the IEEE Conference on Computer Vision and Pattern Recognition (CVPR), 2018.
> >
> > [28] Jifeng Dai, Kaiming He, and Jian Sun. Boxsup: Exploiting bounding boxes to supervise convolutional networks for semantic segmentation. In Proceedings of International Conference on Computer Vision (ICCV), 2015.
> >
> > [29] Anna Khoreva, Rodrigo Benenson, Jan Hosang, Matthias Hein, and Bernt Schiele. Simple does it: Weakly supervised instance and semantic segmentation. In Proceedings of the IEEE Conference on Computer Vision and Pattern Recognition (CVPR), 2017.
> >
> > [30] Seong Joon Oh, Rodrigo Benenson, Anna Khoreva, Zeynep Akata, Mario Fritz, and Bernt Schiele. Exploiting saliency for object segmentation from image level labels. In Proceedings of the IEEE Conference on Computer Vision and Pattern Recognition (CVPR), 2017.
> >
> > [31] Yu Zeng, Yunzhi Zhuge, Huchuan Lu, and Lihe Zhang. Joint learning of saliency detection and weakly supervised semantic segmentation. In Proceedings of International Conference on Computer Vision (ICCV), 2019.

---

> > > ### Comment · Reviewer_RcgA · 2021-08-31
> > > **Re: Thank you for your insightful comments**
> > >
> > > Thank you for your detailed response. Overall my concerns have been addressed, especially regarding the related work. Therefore I will improve my rating of this paper accordingly. In summary I think this is a good paper, however I still see some minor points where this paper could be improved further.
> > >
> > > **Testing on held-out data.**
> > > I appreciate that you agree that "testing on held-out samples should also be the norm for unsupervised methods" and that you mention this more explicitely. I understand that retraining and reevaluating models that were evaluated differently is a tremendous effort, so following previous works in this respect is an adequate decision.
> > > Nevertheless I think this work could push more towards proper evaluation. The main result table (Table 1) could e.g. be structured using the evaluation on held-out data as main result (where available) and contain results from the training data somehow marked as secondary (e.g. using braces).
> > >
> > > **Testing generative modeling capabilities.**
> > > I appreciate that you're going to show preliminary results of sampling from the model. I still think it would have been interesting to go more into depth regarding the generative modeling capabilities of the model, not only to create realistic samples but also to obtain more insights into the inner workings of the model. Since the main focus of this work is foreground-background segmentation, leaving this for future work as you plan is however justified.

---

### Official Review · Reviewer_GuBW · 2021-07-17

**Rating:** 7
**Confidence:** 3

**Summary:**

This work proposes a novel unsupervised foreground segmentation technique based on Mixture of Experts energy based modelling with a generative neural network. They introduce an interesting inductive bias, pixel reassignment, which works as the main regularizer and the secret sauce that enables the fg/bg separation. They essentially reshuffle the pixels for the background generation based on the learned index matrix and ablations shows that's the most significant regularizer.
They compare their results on 6 datasets, including birds, dogs, and clevr6 to both object centric approaches such as slot attention and gan based methods. They consistently outperform the previous methods with a significant margin.

**Limitations And Societal Impact:**

The failure cases are mentioned in the appendix briefly. One question is the computation cost and speed for training and inference.

**Main Review:**

This work formulates the foreground/background separation as a 3 part task. The latent space prior which they use an energy based formulation LEBM. The pixel assignment classification. The generative model based on the assignment and the latent. They use a langevin dynamics mcmc to iteratively refine their image separation. They also add the ability to reshuffle the pixels for background generation, intuitively make it more coherent. Moreover, they augment the latent z in their LEBM formulation with a one hot vector y which later on they can use as pseudo labels for enforcing the bipartition. Based on the ablation studies the pseudo label and the pixel reassignment are the two most crucial manipulations for this method to work.

Apart from impressive results on 6 datasets and outperforming the prior work, they also evaluate their model on generalizing to unseen categories and novel objects. The results suggests that their model learns a consistent model of the background pixels which is to some degree robust to new objects. (Table 4 results should be bold for best in the middle row of each category).

The writeup is quite dense with too many notations but the additional info in the appendix provide enough context. The pixel reassignment is not clear when is applied. Maybe adding a line to the algorithm or fig.1 would help clarifying this.

**Time Spent Reviewing:**

3

---

> ### Author Response · Authors · 2021-08-10
> **Thank you for your insightful comments**
>
> We sincerely thank you for your kind words and thoughtful comments. Below, we provide point-to-point responses to address the questions.
>
> **Computation cost and speed for training and inference**
>
> As mentioned in supplemental material B L71-73, we run experiments on a single V100 GPU with 16GB RAM and with a batch size of 48. We set the maximum training iterations to 10K and run for at most 48 hours for each dataset. Although our method involving MCMC sampling is more costly than the encoder networks in Slot Attention models, it is comparable to the amortized iterative inference in IODINE.
>
> **Adding a line to the algorithm or fig.1 to help clarify pixel re-assignment**
>
> Thanks for your advice. We would modify the illustration as suggested. We have provided pytorch-style code in the supplemental material D L134-154 to facilitate understanding. In our implementation, the re-assignment function follows the output of a decoder network, which is a shuffling grid with the same size of the image. Its values indicate the "re-assigned" pixel coordinates. We aggregate the original background image pixels using these re-assigned pixel coordinates to perform the pixel re-assignment.
>
> **Formatting suggestion**
>
> >Table 4 results should be bold for best in the middle row of each category
>
> We would follow your suggestion to revise the paper. Thank you.

---

### Official Review · Reviewer_QLw6 · 2021-07-19

**Rating:** 6
**Confidence:** 3

**Summary:**

The paper tackles the problem of unsupervised foreground segmentation with a combination of energy-based and deep-learning-based models. It models the foreground and background as two components that can be selected by a latent variable. The background has an inductive bias that pixels can be re-assigned to a different location. The whole model is generative. During inference, EM is needed to figure out the foreground and background separation by maximizing probability to generate the image and set the right latent states.

**Ethics Review Area:**

["I don’t know"]

**Limitations And Societal Impact:**

It is a general purpose method. Not sure if there is any specific societal impact.

**Main Review:**

Pros:
1.	The method is built on decent math. With the factorized probabilistic formulation, the optimization follows naturally by applying established techniques.
2.	The idea of using pixel reassignment as an inductive bias is interesting though used in the previous non-deep-learning method.
3.	The experimental results demonstrate significant performance improvement over a few previous methods for foreground-background separation

Cons:
1.	It is unclear if the proposed method is scalable. What are the sizes of the images used in experiments? Can the method achieve similar relative performance gain over other methods on higher resolution images? What’s the method’s efficiency compared to other methods?
2.	There is a line of work to automatically discover object structures, such as landmarks and object parts. For example, a CVPR’19 paper, “Unsupervised Part-Based Disentangling of Object Shape and Appearance,” (not cited. Just taking this as an example. There are more papers) can discover object parts, which can be adapted to foreground detection. Compared to this line of work, what is the advantage of the proposed method? And is there technical connections?

---

Thanks for the authors' response.

**Time Spent Reviewing:**

3

---

> ### Author Response · Authors · 2021-08-10
> **Thank you for your insightful comments**
>
> We sincerely thank you for your time and constructive comments. Below, we provide point-to-point replies to your comments in order and hopefully resolve the remaining questions you have.
>
> **Input image size and scalability of the method**
>
> We follow the previous methods and set the input image size as $128\times 128$. Please refer to supplemental material A for more details about the data curation. We agree that it would be interesting to examine the performance of the baselines and the proposed method on images with higher resolutions. However, as most of the baselines are purposely designed for the fixed resolution of $128\times 128$, we directly follow this convention; it would be difficult to compare the performances with higher resolutions across all methods. We would like to leave it for future research. In theory, we see no obvious issues in scaling up the proposed method for higher resolution, though the complexity is yet to be investigated.
>
> **The method's efficiency compared to other methods**
>
> As mentioned in supplemental material B L71-73, we run experiments on a single V100 GPU with 16GB RAM and with a batch size of 48. We set the maximum training iterations to 10K and run for at most 48 hours for each dataset. Although our method involving MCMC sampling is more costly than the encoder networks in Slot Attention models, it is comparable to the amortized iterative inference in IODINE.
>
> **Connection with a line of work that automatically discovers object structures**
>
> Thanks for pointing us to this line of work. We would cite related papers and make connections in revision. We agree that automatically discovering object structures, such as landmarks and object parts, has a close connection with foreground extraction. The mentioned method [1] is, however, not directly comparable with the proposed method as the target task and experiment setups are different. Specifically, [1] aims to learn a disentangled part-level representation of object shape and appearance, whereas the target task of the proposed method is to extract the foreground objects from the images. Though it is possible that the part-level shape representation can be transformed into the object-level foreground mask, the exact performance of [1] on unsupervised foreground extraction remains unknown.
>
> We summarize the possible advantages of our method compared with [1] as follows:
> - In terms of modeling, our method requires less prior knowledge of the foreground objects and is, therefore, more generic. In comparison, the method mentioned in [1], to our understanding, needs to specify the number of object parts when discovering the object structures. Using a different number of object parts may lead to different performances.
> - It is unclear if [1] can handle images with multiple foreground objects, especially if those objects are simple geometries (such as triangles, circles, and squares in TM-dSprites) without explicit part-level features.
> - Since [1] learns a highly specified object representation for each object category, it is possible that the model can perform worse on cross-category generalization. In contrast, our method has shown the generalizability in Table 4.
>
> We believe that the proposed method can potentially benefit from part-level modeling of the foreground object when dealing with challenging cases mentioned in the supplemental material F.2. Taken together, the disentanglement of appearance and shape mentioned in [1] is an interesting topic to consider in future research.
>
> [1] Dominik Lorenz, Leonard Bereska, Timo Milbich, and Bjorn Ommer. Unsupervised part-based disentangling of object shape and appearance. In Proceedings of the IEEE Conference on Computer Vision and Pattern Recognition (CVPR), 2019.

---

### Decision · Program_Chairs · 2021-09-27

**Decision:**

Accept (Poster)

**Comment:**

All reviewers rate this work as interesting and unanimously recommend acceptance of the paper but still see room for improvement.